# Placental genomics mediates genetic associations with complex health traits and disease

Arjun Bhattacharya [1,2✉], Anastasia N. Freedman [3], Vennela Avula[3], Rebeca Harris [4], Weifang Liu [5], Calvin Pan [6], Aldons J. Lusis [6,7,8], Robert M. Joseph[9], Lisa Smeester [3,10,11], Hadley J. Hartwell[3], Karl C. K. Kuban[12], Carmen J. Marsit [13], Yun Li[5,14,15], T. Michael O'Shea[16], Rebecca C. Fry[3,10,11,17✉] & Hudson P. Santos Jr[4,10,17✉]

As the master regulator *in utero*, the placenta is core to the Developmental Origins of Health and Disease (DOHaD) hypothesis but is historically understudied. To identify placental gene-trait associations (GTAs) across the life course, we perform distal mediator-enriched transcriptome-wide association studies (TWAS) for 40 traits, integrating placental multi-omics from the Extremely Low Gestational Age Newborn Study. At $P < 2.5 \times 10^{-6}$, we detect 248 GTAs, mostly for neonatal and metabolic traits, across 176 genes, enriched for cell growth and immunological pathways. In aggregate, genetic effects mediated by placental expression significantly explain 4 early-life traits but no later-in-life traits. 89 GTAs show significant mediation through distal genetic variants, identifying hypotheses for distal regulation of GTAs. Investigation of one hypothesis in human placenta-derived choriocarcinoma cells reveal that knockdown of mediator gene *EPS15* upregulates predicted targets *SPATA13* and *FAM214A*, both associated with waist-hip ratio in TWAS, and multiple genes involved in metabolic pathways. These results suggest profound health impacts of placental genomic regulation in developmental programming across the life course.

[1] Department of Pathology and Laboratory Medicine, David Geffen School of Medicine, University of California, Los Angeles, CA 90095, USA. [2] Institute for Quantitative and Computational Biosciences, David Geffen School of Medicine, University of California, Los Angeles, CA 90095, USA. [3] Department of Environmental Sciences and Engineering, Gillings School of Global Public Health, University of North Carolina, Chapel Hill, NC 27514, USA. [4] Biobehavioral Laboratory, School of Nursing, University of North Carolina, Chapel Hill, NC 27514, USA. [5] Department of Biostatistics, Gillings School of Global Public Health, University of North Carolina, Chapel Hill, NC 27514, USA. [6] Department of Human Genetics, David Geffen School of Medicine, University of California, Los Angeles, CA 90095, USA. [7] Department of Medicine, David Geffen School of Medicine, University of California, Los Angeles, CA 90095, USA. [8] Department of Microbiology, Immunology and Molecular Genetics, David Geffen School of Medicine, University of California, Los Angeles, CA 90095, USA. [9] Department of Anatomy and Neurobiology, Boston University School of Medicine, Boston, MA 02118, USA. [10] Institute for Environmental Health Solutions, Gillings School of Global Public Health, University of North Carolina, Chapel Hill, NC 27514, USA. [11] Curriculum in Toxicology and Environmental Medicine, University of North Carolina, Chapel Hill, NC 27514, USA. [12] Department of Pediatrics, Division of Pediatric Neurology, Boston University Medical Center, Boston, MA 02118, USA. [13] Gangarosa Department of Environmental Health, Rollins School of Public Health Emory University, Atlanta, GA 30322, USA. [14] Department of Genetics, University of North Carolina, Chapel Hill, NC 27514, USA. [15] Department of Computer Science, University of North Carolina, Chapel Hill, NC 27514, USA. [16] Department of Pediatrics, School of Medicine, University of North Carolina, Chapel Hill, NC 27514, USA. [17] These authors contributed equally: Rebecca C. Fry, Hudson P. Santos Jr. ✉email: abtbhatt@ucla.edu; rfry@email.unc.edu; hsantosj@email.unc.edu

The placenta serves as the master regulator of the intrauterine environment via nutrient transfer, metabolism, gas exchange, neuroendocrine signaling, growth hormone production, and immunologic surveillance[1–3]. Owing to strong influences on postnatal health, the placenta is central to the Developmental Origins of Health and Disease (DOHaD) hypothesis—that the in utero experience has lifelong impacts on child health by altering developmental programming and influencing risk of common, non-communicable health conditions[4]. For example, physiological characteristics of the placenta have been linked to neuropsychiatric, developmental, and metabolic diseases or health traits (collectively referred to as traits) that manifest throughout the life course, either early- or later-in-life (Fig. 1)[1,5–8]. Despite its long-lasting influences on health, the placenta is understudied in large consortia studies of multi-tissue gene regulation[9,10]. Studying regulatory mechanisms in the placenta underlying biological processes in developmental programming could provide novel insight into health and disease etiology.

The complex interplay between genetics and placental transcriptomics and epigenomics has strong effects on gene expression that may explain variation in gene-trait associations (GTAs). Quantitative trait loci (QTL) analyses have identified a strong influence of *cis*-genetic variants on both placental gene expression and DNA methylation[11]. Furthermore, there is growing evidence that the placental epigenome influences gene regulation, often distally (>1–3 Megabases away in the genome)[12], and that placental DNA methylation and microRNA (miRNA) expression are associated with health traits in children[13]. Dysfunction of transcription factor regulation in the placenta has also shown profound effects on childhood traits[14]. Although combining genetics, transcriptomics, and epigenomics lends insight into the influence of placental genomics on complex traits[15], genome-wide screens for GTAs that integrate different molecular profiles and generate functional hypotheses require more sophisticated computational methods.

To this end, advances in transcriptome-wide association studies (TWAS) have allowed for integration of genome-wide association studies (GWAS) and eQTL datasets to boost power in identifying GTAs, specific to a relevant tissue[16,17]. However, traditional methods for TWAS largely overlook genetic variants distal to genes of interest, ostensibly mediated through regulatory biomarkers, like transcription factors, miRNAs, or DNA methylation sites[18]. Not only may these distal biomarkers explain a significant portion of both gene expression heritability and trait heritability on the tissue-specific expression level[19,20], they may also influence tissue-specific trait associations for individual genes. Owing to the strong interplay of regulatory elements in placental gene regulation, we sought to systematically characterize portions of gene expression that are influenced by these distal regulatory elements.

Here, we set out to identify the following: (1) which genes show associations between their placental genetically regulated expression (GReX) and various traits across the life course, (2) which traits along the life course can be explained by placental GReX, in aggregate, and (3) which transcription factors, miRNAs, or CpG sites potentially regulate trait-associated genes in the placenta (Fig. 1). We leveraged multi-omic data from fetal-side placenta tissue from the Extremely Low Gestational Age Newborn (ELGAN) Cohort Study[21] to train predictive models of gene expression enriched for distal-SNPs using MOSTWAS, a recent TWAS extension that integrates multi-omic data[22]. Using 40 GWAS of European-ancestry subjects from large consortia[23–27], we performed a series of TWAS for non-communicable health traits and disorders that may be influenced by the placenta to identify GTAs and functional hypotheses for regulation (Fig. 2). To our knowledge, this analysis is among the first distal mediator-enriched TWAS of health traits that integrates multi-omic data from the placenta.

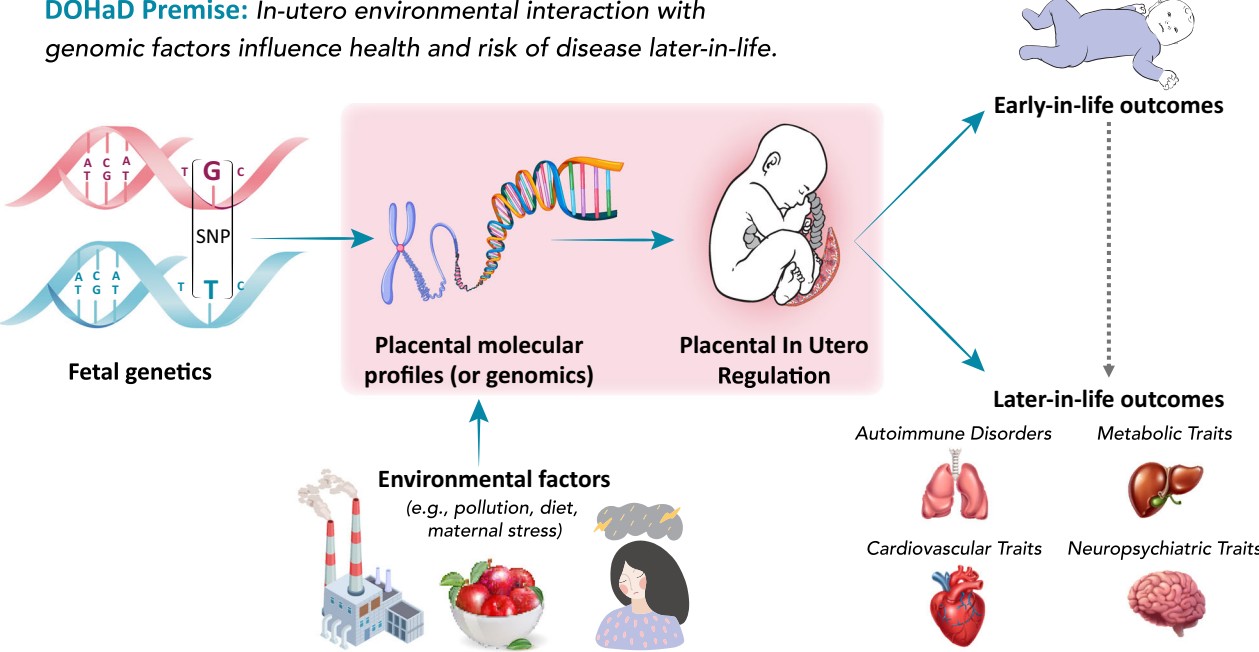

**Fig. 1 Overview of the DOHaD hypothesis.** The placenta facilitates important functions in utero, including nutrient transfer, metabolism, gas exchange, neuroendocrine signaling, growth hormone production, and immunologic control. Accordingly, it is a master regulator of the intrauterine environment and is core to the Developmental Origins of Health and Disease (DOHaD) hypothesis. Placental genomic regulation is influenced by both genetic and environmental factors and affects placental developmental programming. In turn, this programming has been shown to have profound impacts on a variety of disorders and traits, both early- and later-in-life.

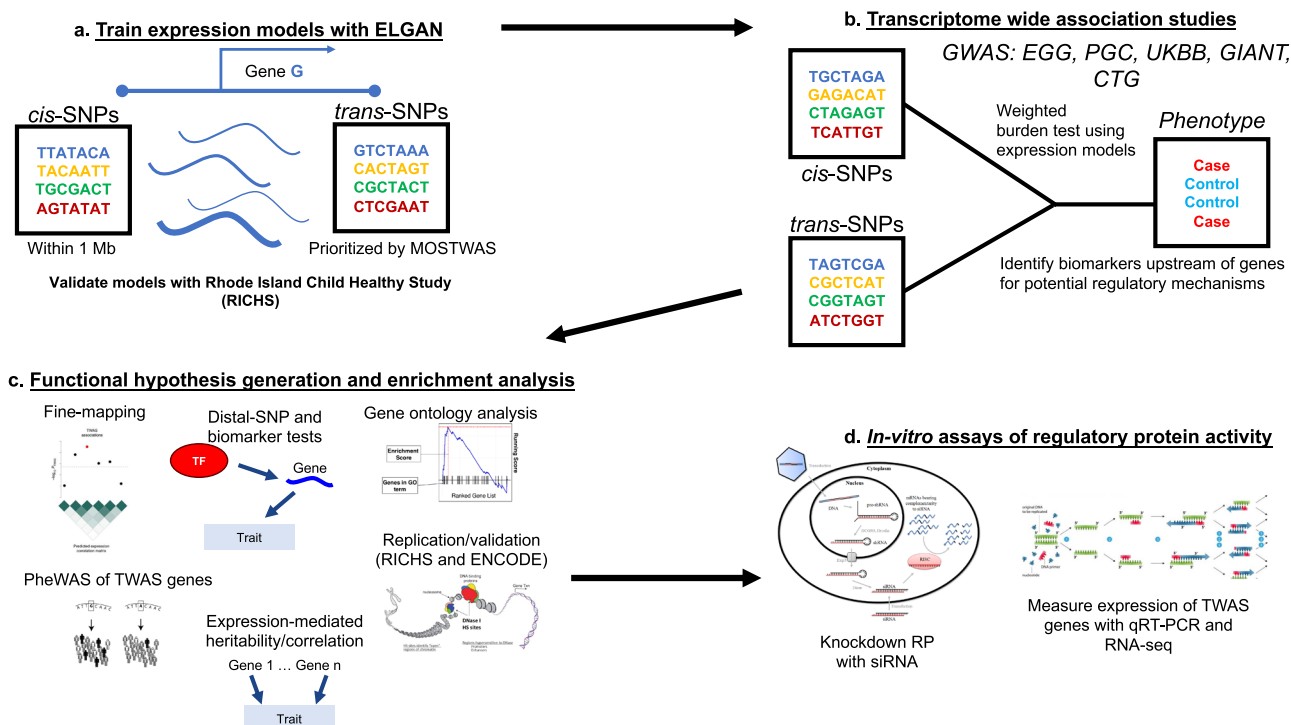

**Fig. 2 Overview of analytic pipeline. a** Predictive models of placental expression are trained from germline genetics, enriched for mediating biomarkers using MOSTWAS[22] and externally validated in RICHS[36] eQTL data. **b** Predictive models are integrated with GWAS for 40 traits to detect placental gene-trait associations (GTAs). **c** GTAs are followed up with gene ontology analyses, probabilistic fine-mapping, and phenome-wide scans of genes with multiple GTAs. Relationships between identified distal mediators and TWAS genes are investigated further in RICHS and ENCODE[10]. Expression-mediated genetic heritability of and correlations between traits are estimated. **d** In vitro validation of prioritized regulatory protein-TWAS gene pairs are conducted using placenta-derived choriocarcinoma cells by gene silencing, qRT-PCR, and RNA-seq to measure transcriptomic impacts of the regulatory protein.

## Results

**Overview of analytic framework**. We conduct a series of distal mediator-enriched transcriptome-wide association studies (TWAS) for a variety of complex traits by integrating GWAS data with placental eQTL data from ELGAN. First, we use a recent methodology, MOSTWAS[22], to train predictive models of gene expression using both local- and distal-SNPs to genes (Fig. 2a). Next, we employ these models to conduct TWAS for these traits using GWAS summary statistics to identify genes with placental genetically regulated expression (GReX) associated with different traits across the life course (Fig. 2b)[17]. We then estimate the extent to which placental genetically regulated expression across all trait-associated genes can explain the variability in a trait and correlations between traits (Fig. 2c)[17,28]. Next, to provide more biological context, for genes estimated to have placental GTAs, we run multiple follow-up analyses (Fig. 2c): gene ontology enrichment analyses[29], probabilistic fine-mapping of overlapping loci[30], phenome-wide analyses for select genes, and prioritization of functional hypotheses for upstream distal regulation[22]. Lastly, for one particular functional hypothesis supported with strong computational evidence, we conduct an in vitro assay in a human placenta-derived cell line to validate the predicted mediator-TWAS gene relationship and the transcriptomic consequences of this mediator (Fig. 2d).

**Complex traits are genetically heritable and correlated**. We curated GWAS summary statistics from subjects of European ancestry for 40 non-communicable traits and disorders across five health categories to identify potential links to genetically regulated placental expression (traits and cohorts for each GWAS are summarized in Supplementary Data 1, sample sizes are provided

in Supplementary Data 2). These five categories of traits (autoimmune/autoreactive disorders, metabolic traits, cardiovascular disorders, early childhood outcomes, and neuropsychiatric traits) have been linked previously to placental and fetal biology and morphology[1–8]. These 40 traits, derived from 5 different consortia (Supplementary Data 1), comprise of 3 autoimmune/autoreactive disorders, 8 body size/metabolic traits, 4 cardiovascular disorders, 14 neonatal/early childhood traits, and 11 neuropsychiatric traits/disorders[23–27]. The 26 traits that are not categorized as neonatal/early childhood traits are measured exclusively in adults. In addition, these 40 GWAS are not derived from the same samples of patients.

To quantify the total genetic contribution to each trait and the genetic associations shared between traits, using linkage disequilibrium (LD) score regression with LD scores generated for individuals of European ancestry from the 1000 Genomes projects[31,32], we estimated the SNP heritability ($h^2$) and genetic correlation ($r_g$) of these traits, respectively (Supplemental Figs. S1 and S2). Of the 40 traits, 37 showed significantly positive SNP heritability and 18 with $\hat{h}^2 > 0.10$ (Supplemental Fig. S1, Supplementary Data 1), with the largest heritability for childhood BMI ($\hat{h}^2 = 0.69, \text{SE} = 0.064$). As expected, we observed strong, statistically significant genetic correlations between traits of similar categories (i.e., between neuropsychiatric traits or between metabolic traits) (Supplemental Fig. S2; Supplementary Data 3). At Benjamini–Hochberg FDR-adjusted $P < 0.05$, we also observed significant correlations between traits from different categories: diabetes and angina ($\hat{r}_g = 0.51$, FDR-adjusted $P = 6.53 \times 10^{-33}$), Tanner scale (in children) and BMI ($\hat{r}_g = 0.42$, FDR-adjusted $P = 1.06 \times 10^{-3}$), and BMI and obsessive compulsive disorder

($\hat{r}_g = -0.28$, FDR-adjusted $P = 1.79 \times 10^{-9}$), for example. Given strong and potentially shared genetic influences across these traits, we examined whether genetic associations with these traits are mediated by the placental transcriptome.

**Multiple placental gene-trait associations detected across the life course**. In the first step of our TWAS (Fig. 2a), we leveraged MOSTWAS[22], a recent TWAS extension that includes distal variants in transcriptomic prediction, to train predictive models of placental expression. As large proportions of total heritable gene expression are explained by distal-eQTLs local to regulatory hotspots[18,20], MOSTWAS uses data-driven approaches to identify mediating regulatory biomarkers or distal-eQTLs mediated through local regulatory biomarkers to increase predictive power for gene expression and power to detect GTAs (Supplemental Fig. S3)[22]. In this analysis, these regulatory biomarkers include potential regulatory protein (RP) encoding genes (as curated by TFcheckpoint[33]), miRNAs, and CpG methylation sites from the ELGAN Study. we assume that these RP genes, miRNAs, and genes and other regulatory features local to these CpG methylation sites have distal effects on the transcription of genes of interest and thus potentially mediate distal-eQTLs to the gene of interest (Methods).

Using genotypes from umbilical cord blood[34] and mRNA expression, CpG methylation, and miRNA expression data from fetal-side placenta[15] from the ELGAN Study[21] for 272 infants born pre-term, we built genetic models to predict RNA expression levels for genes in the fetal placenta (demographic summary in Supplementary Data 4). Out of a total of 12,020 genes expressed across all samples in ELGAN, we successfully built significant models for 2994 genes, with positive SNP-based expression heritability (nominal $P<0.05$) and fivefold McNemar's adjusted cross-validation (CV) $R^2 \geq 0.01$ (Fig. 3a [Step 1]; Methods). Only these 2,994 models are used in subsequent TWAS steps. Mean SNP heritability for these genes was 0.39 (25% quantile = 0.253, 75% quantile = 0.511), and mean CV $R^2$ was 0.031 (quantiles: 0.014, 0.034). For out-sample validation, we imputed expression into individual-level genotypes from the Rhode Island Child Health Study (RICHS; $N = 149$)[35,36], showing strong portability across studies: of 2,005 genes with RNA-seq expression in RICHS, 1,131 genes met adjusted $R^2 \geq 0.01$, with mean $R^2$ of 0.011 (quantiles: $7.71 \times 10^{-4}$, 0.016) (Fig. 3b and Supplementary Data 5). Summary statistics of demographic and clinical variables for the RICHS show similar distributions of race, though RICHS excluded all pre-term babies, a clear difference in these two cohorts (Supplementary Data 4).

We integrated GWAS summary statistics for 40 traits from European-ancestry subjects with placental gene expression using our predictive models. Using the weighted burden test with the 1000Genomes European ancestry LD matrix as a reference[17], we detected 932 GTAs (spanning 686 unique genes) at $P<2.5 \times 10^{-6}$, a transcriptome-wide significance threshold consistent with previous TWAS[17,28] (Fig. 3a [Step 2]). As many of these loci carry significant signal because of strong SNP-trait associations, we employed Gusev et al.'s permutation test to assess how much signal is added by the SNP-expression weights and confidently conclude that integration of expression data significantly refines association with the trait[17]. At FDR-adjusted $P<0.05$ and spanning 176 unique genes, we detected 248 such GTAs, with 11 autoimmune/autoreactive, 136 body size/metabolic, 32 cardiovascular, 39 neonatal/childhood, and 30 neuropsychiatric GTAs (Fig. 3a [Step 3], Supplementary Data 2 and 6; Miami plots of TWAS Z-scores in Supplemental Figs. S4–S9).

The 39 GTAs detected with adult BMI included *LARS2* ($Z = 11.4$) and *CAST* ($Z = -4.61$). These two GTAs have been

detected using *cis*-only TWAS in different tissues[17,28]. In addition, one of the 30 genes identified in association with waist-hip ratio (in adults) was prioritized in other tissues by TWAS: *NDUFS1* ($Z = -5.38$)[28]. We cross-referenced susceptibility genes with a recent *cis*-only TWAS of fetal birthweight, childhood obesity, and childhood BMI by Peng et al. using placental expression data from RICHS[8]. Of the 19 birthweight-associated genes they identified, we could only train significant expression models for two in ELGAN: *PLEKHA1* and *PSG8*. We only detected a significant association between *PSG8* and fetal birthweight ($Z = -7.77$). Similarly, of the six childhood BMI-associated genes identified by Peng et al., only 1 had a significant model in ELGAN and showed no association with the trait; there were no overlaps with childhood obesity-associated genes[8]. We hypothesize that minimal overlap with susceptibility genes identified by Peng et al. is due to differing phenotypes and eQTL architectures in the datasets and different inclusion criteria for significant gene expression models.

Next, we tested for horizontal pleiotropic effects of the SNPs employed in the models for TWAS-prioritized genes; if SNPs affect the outcome through a pathway independent of expression of the gene, the TWAS association may be biased[37,38]. Here, using PMR-Summary-Egger[38], we test the magnitude of this null hypothesis for each of the 248 TWAS-prioritized GTAs. At FDR-adjusted $P<0.05$, only three GTAs showed significant horizontal pleiotropic effects: *MOV10*, *SLC35G2*, and *HLA-A*, all associated with adult waist-hip ratio (Supplementary Data 6). These three genes may have upwardly biased TWAS associations, as the SNPs used to construct their GReX may influence the outcome through a different molecular pathway.

As these GTAs indicate trait association and do not reflect causality, we used FOCUS[30], a Bayesian gene-level fine-mapping approach. For TWAS-significant genes with overlapping genetic loci, FOCUS estimates posterior inclusion probabilities (PIP) in a credible set of genes that explains the association signal at the locus. We found 8 such overlaps and estimated a 90% credible set of genes explaining the signal for each locus (Supplementary Data 9). For example, we identified 3 genes associated with triglycerides in adults at the 12q24.13 chromosomal region (*ERP29*, *RPL6*, *BRAP*), with *ERP29* defining the region's 90% credible set with approximately 95% PIP. Similarly, we detected 3 genes associated with adult BMI at 10q22.2 (*AP3M1*, *SAMD8*, *MRPS16*), with *AP3M1* defining the region's 90% credible set with approximately 99% PIP.

We conducted over-representation analysis for biological process, molecular function, and PANTHER gene pathway ontologies for TWAS-detected susceptibility genes (Fig. 3d and Supplementary Data 7)[29]. Overall, considering all 176 TWAS-identified genes, we observed enrichments for nucleic acid binding and immune or cell growth signaling pathways (e.g., B-cell/T-cell activation and EGF receptor, interleukin, PDGF, and Ras signaling pathways). By trait, we found related pathways (sphingolipid biosynthesis, cell motility, etc) for TWAS genes for metabolic and morphological traits (e.g., BMI and childhood BMI); for most traits, we were underpowered to detect ontology enrichments. We also assessed the overlap of TWAS genes with GWAS signals. A total of 112 TWAS genes did not overlap with GWAS loci ($P<5 \times 10^{-8}$) within a 500 kilobase interval around any SNPs (local and distal) included in predictive models (Supplementary Data 10).

**Genetically regulated placental expression mediates trait heritability and genetic correlations**. To assess how genetically regulated placental expression explains trait variance, we computed trait heritability on the placental expression level ($h^2_{GE}$) using all examined and all TWAS-prioritized susceptibility genes

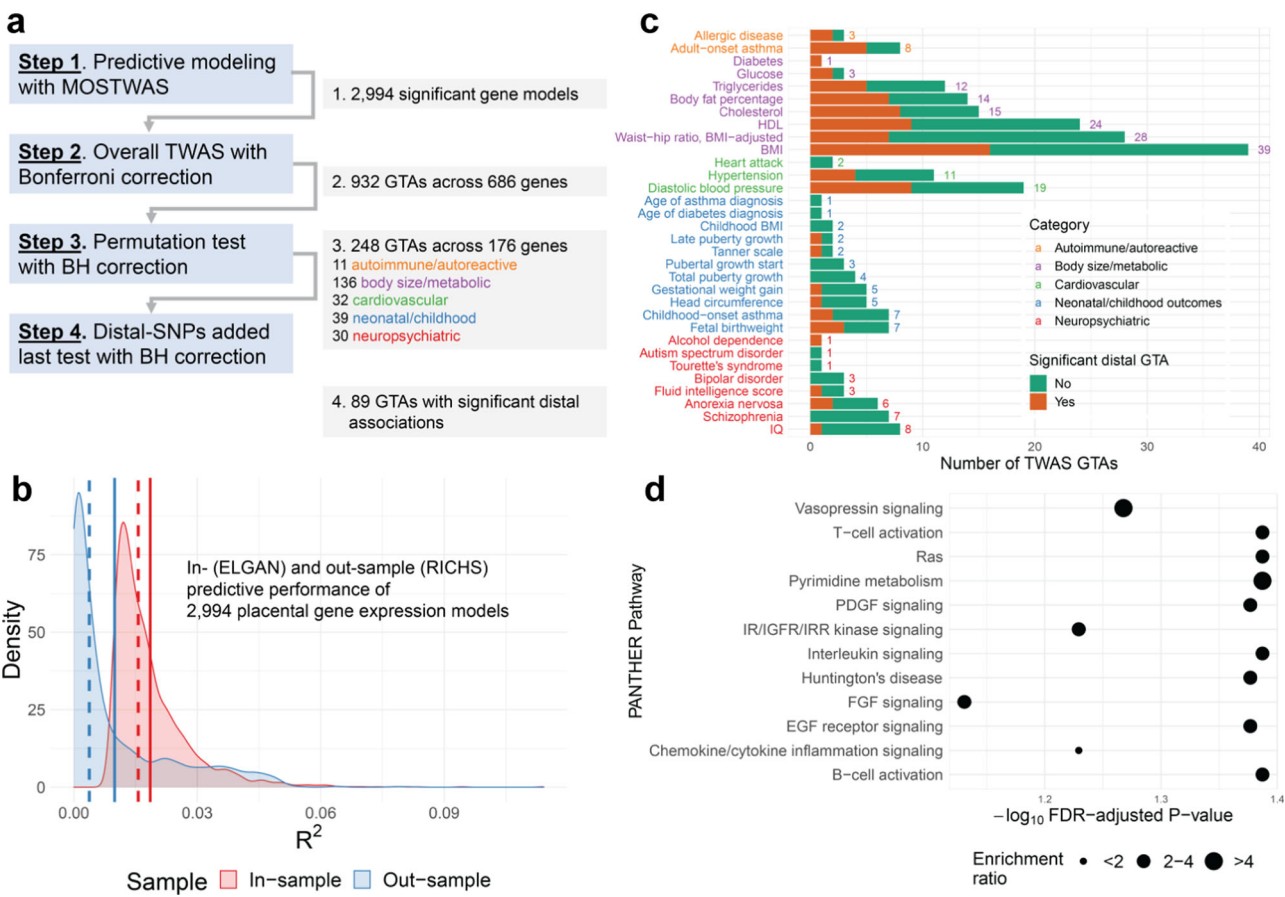

**Fig. 3 Placental MOSTWAS prediction and association test results. a** Overview of TWAS association testing pipeline with number of gene-trait associations (GTAs) across unique genes over various levels of TWAS tests. **b** Kernel density plots of in- (through cross-validation in ELGAN, red) and out-sample (external validation in RICHS, blue) McNemar's adjusted $R^2$ between predicted and observed expression. Dotted and solid lines represent the mean and median of the respective distribution, respectively. **c** Bar graph of numbers of TWAS GTAs at overall TWAS $P<2.5\times10^{-6}$ (for two-sided weighted burden $Z$-test) and two-sided permutation test FDR-adjusted $P<0.05$ (x-axis) across traits (y-axis). The total number of GTAs per trait are labeled, colored by the category of each trait. The bar is broken down by numbers of GTAs with (orange) and without (green) significant distal expression-mediated associations, as indicated by FDR-adjusted $P<0.05$ for the distal-SNPs added-last test. **d** Enrichment plot of over-representation in 176 TWAS genes of PANTHER pathways (y-axis) with -$\log_{10}$ FDR-adjusted $P$-value (x-axis) of one-sided Fisher's exact test. The size of the point gives the relative enrichment ratio for the given pathway.

using a LD-score regression approach[17,31]. Overall, we found 4/14 neonatal traits (childhood BMI, head circumference, total puberty growth, and pubertal growth start) with significant $\hat{h}_{GE}^2 > 0$ (FDR-adjusted $P<0.05$ for jack-knife test of significance)[28]; none of the 26 traits outside the neonatal category were appreciably explained by placental GReX (Supplemental Fig. S10). Figure 4a shows that mean $\hat{h}_{GE}^2$ is higher in neonatal traits than other groups. In fact, placenta expression-mediated genetic heritability explains a larger proportion of total SNP heritability of neonatal traits, compared to traits from other categories (Fig. 4b). A comparison of the number of GWAS-significant SNPs and TWAS-significant genes also shows that neonatal traits are enriched for placental TWAS associations, even though significant genome-wide GWAS architecture cannot be inferred for these traits (Supplemental Fig. S11). These observations suggest that placental GReX affects neonatal traits more profoundly, as a significantly larger proportion of neonatal traits showed significant heritability on the placental GReX level than later-in-life traits.

Using RHOGE[28], we assessed genetic correlations ($r_{GE}$) between traits at the level of placental GReX (Supplemental Fig. S12). We found several known correlations: between cholesterol and triglycerides, both in adults, ($\hat{r}_{GE} = 0.99, P = 1.44\times10^{-118}$) and

childhood BMI and adult BMI ($\hat{r}_{GE} = 0.55, P = 3.67\times10^{-8}$). Interestingly, we found correlations between traits across categories (Fig. 4c): IQ and diastolic blood pressure, both in adults, ($\hat{\rho}_{GE} = -0.55, P = 2.44\times10^{-5}$) and age of asthma diagnosis and adult glucose levels ($\hat{\rho}_{GE} = 0.86, P = 3.05\times10^{-6}$). These traits have been linked in morphological analyses of the placenta, but our results suggest possible genomic contributions[39]. Overall, these correlations suggest shared genetic pathways for these pairs of traits or for etiologic antecedents of these traits; these shared pathways could be either at the susceptibility genes or through shared distal loci, mediated by RPs, miRNAs, or CpG methylation sites.

**Genes with multiple GTAs have phenome-wide associations in early- and later-life traits.** We noticed that multiple genes were identified in GTAs with multiple traits, leading us to examine potential horizontally pleiotropic genes. Of the 176 TWAS-prioritized genes, we identified 50 genes associated with multiple traits, many of which are genetically correlated (Supplementary Data 11). Nine genes showed >3 GTAs across different categories. For example, *IDI1*, a gene involved in cholesterol biosynthesis[40], showed associations with three metabolic and two neuropsychiatric traits: body fat percentage ($Z = 15.57$), HDL

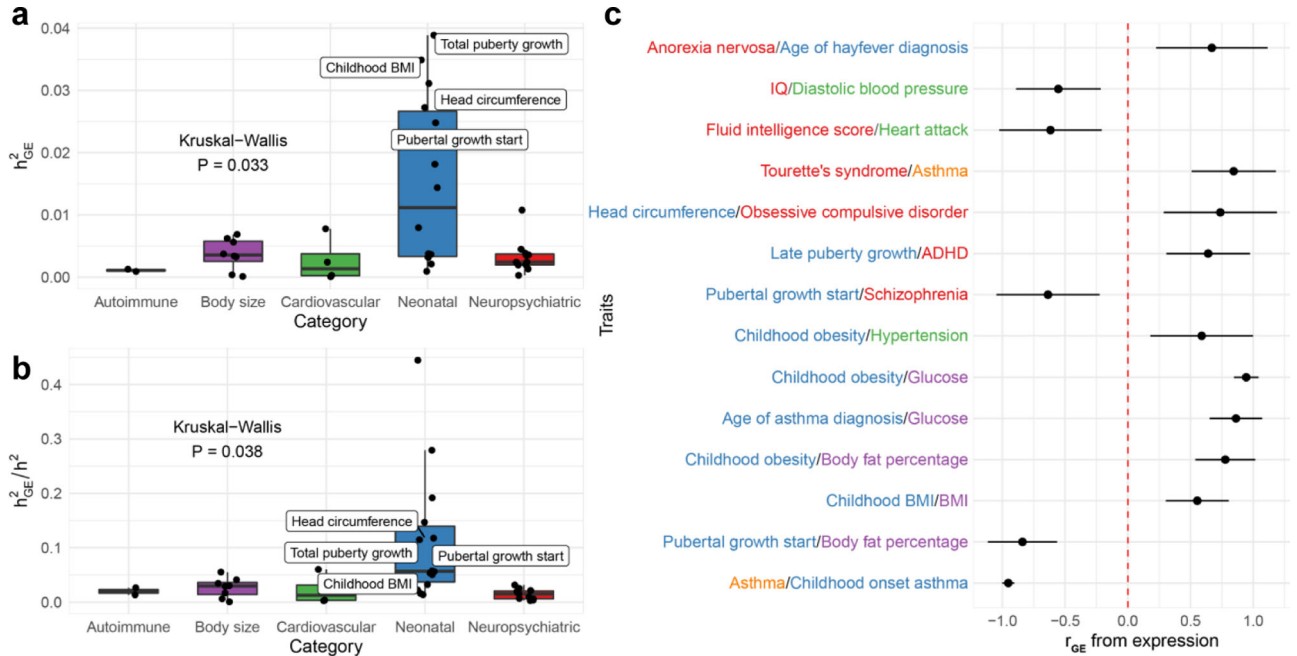

**Fig. 4 Trait genetic heritability and correlations mediated by placental expression. a** Box-plot of expression-mediated trait heritability ($h^2_{GE}$), estimated with LD-score regression, (y-axis) by category (x-axis), with labels if $\hat{h}^2_{GE}$ is significantly greater than 0 using two-sided jack-knife test of significance. Boxplots are **b** Box-plot of expression-mediated trait heritability ($h^2_{GE}$) standardized by SNP heritability ($h^2$) (y-axis) by category (x-axis), with labels if $\hat{h}^2_{GE}$ is significantly greater than 0 using two-sided jack-knife test of significance. **c** Forest plot of significant placenta expression-mediated genetic correlations (shown as points) and 95% FDR-adjusted two-sided Wald-type confidence intervals between traits from different categories. Boxplots in **a**, **b** provide, in order of bottom to top, the lower extreme, lower quartile, median, upper quartile, and upper extreme. Sample sizes used to derive these statistics across **a–c** are provided in Supplementary Data 1.

($Z = 26.48$), triglycerides ($Z = -7.53$), fluid intelligence score ($Z = 6.37$), and schizophrenia ($Z = -5.56$), with all five traits measured in adults. A link between cholesterol-related genes and schizophrenia has been detected previously, potentially due to coregulation of myelin-related genes[41]. Mediated by CpG site cg01687878 (found within *PITPNM2*), predicted expression of *IDI1* was also computed using distal-SNPs within Chromosome 12q24.31, a known GWAS risk loci for hypercholesteremia[42]; the inclusion of this locus may have contributed to the large TWAS associations. Similarly, *SAMD4A* also shows associations with four adult body size/metabolic—body fat percentage ($Z = 6.70$), cholesterol ($Z = -6.76$), HDL ($Z = -6.78$), triglycerides ($Z = -5.30$)—and one adult cardiovascular trait (diastolic blood pressure with $Z = -5.29$). These associations also pick up on variants in Chromosome 12q24.31 local to CpG sites cg05747134 (within *MMS19*) and cg04523690 (within *SETD1B*). Another gene with multiple trait associations is *CMTM4*, an angiogenesis regulator[43], showing associations with body fat percentage ($Z = 6.17$), hypertension ($Z = 5.24$), and fetal birthweight ($Z = 8.11$). *CMTM4* shows evidenced risk of intrauterine growth restriction due to involvement with endothelial vascularization[44], potentially suggesting that *CMTM4* has a more direct effect in utero, which mediates its associations with body fat percentage and hypertension.

We further studied the nine genes with three or more distinct GTAs across different categories (Fig. 5a). Using UK Biobank[23] GWAS summary statistics, we conducted TWAS for a variety of traits, measured in adults, across eight groups, defined generally around ICD code blocks (Fig. 5a and Supplemental Fig. S13); here, we grouped metabolic and cardiovascular traits into one category for ease of analysis. At FDR-adjusted $P<0.05$, *ATPAF2*, *RPL6*, and *SEC11A* showed GTA enrichments for immune-related traits, *ATAPF2* for neonatal traits, *IDI1* for mental

disorders, and *RPS25* for musculoskeletal traits. Across these 8 trait groups, *RPL6* showed multiple strong associations with circulatory, respiratory, immune-related, and neonatal traits (Fig. 5a). Examining specific GTAs for *ATPAF2*, *IDI1*, *RPS25*, and *SEC11A* reveals associations with multiple biomarker traits (Supplemental Fig. S13). For example, at $P<2.5\times10^{-6}$, *ATPAF2* and *IDI1*'s immune GTA enrichment includes associations with eosinophil, monocyte, and lymphocyte count and IGF-1 concentration. *ATPAF* and *RPS25* show multiple associations with platelet volume and distribution and hematocrit percentage. In addition, *IDI1* was associated with multiple mental disorders (obsessive compulsive disorder, anorexia nervosa, bipolar disorder, and general mood disorders), consistent with its TWAS associations with fluid intelligence and schizophrenia (Supplemental Fig. S13). As placental GReX of these genes correlates with biomarkers, these results may not necessarily signify shared genetic associations across multiple traits. Rather, this may point to more fundamental effects of these TWAS-identified genes that manifest in complex traits later in life.

We next examined whether placental GReX of these nine genes correlate with fundamental traits at birth. We imputed expression into individual-level ELGAN genotypes ($N = 729$). Controlling for race, sex, gestational duration, inflammation of the chorion, and maternal age, as described in Methods, we tested for associations for six representative traits measured at birth or at 24 months: neonatal chronic lung disease, birth head circumference Z-score, fetal growth restriction, birthweight Z-score, necrotizing enterocolitis, and Bayley II Mental Development Index (MDI) at 24 months[15]. Shown in Fig. 5b and Supplementary Data 12, at FDR-adjusted $P<0.05$, we detected negative associations between *SEC11A* GReX and birthweight Z-score (effect size: −0.248, 95% adjusted CI: [−0.434,−0.063]) and GReX of *ATPAF2* and head circumference Z-score (−0.173, [−0.282,−0.064]). Furthermore, we

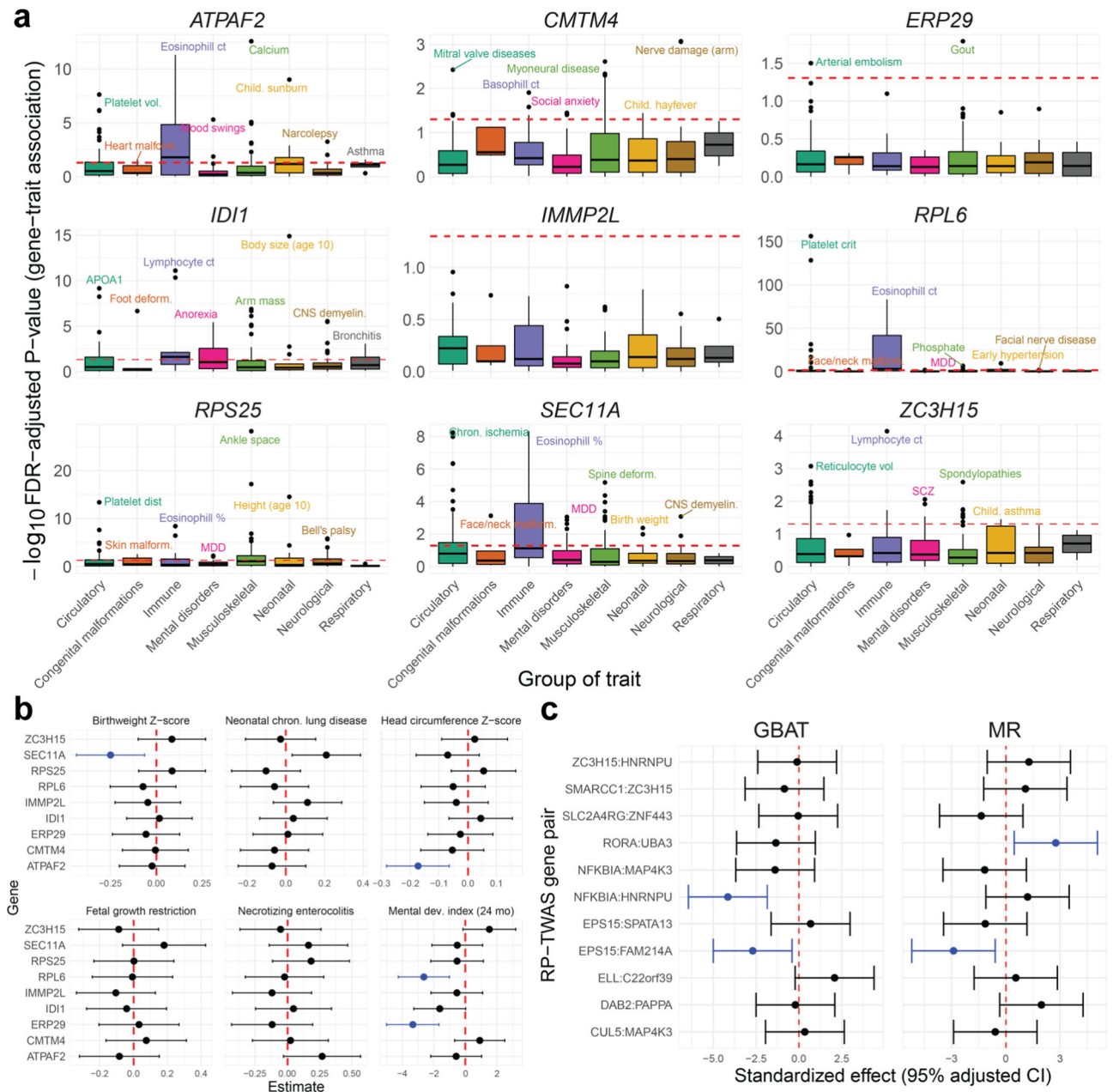

**Fig. 5 Computational follow-up analyses of TWAS-prioritized genes. a** Box-plot of -log10 FDR-adjusted *P*-value (two-sided weighted burden *Z*-tests) of multi-trait scans of GTAs in UKBB, grouped by eight groups of traits (grouped generally around ICD code blocks or organ systems) across nine genes with multiple TWAS GTAs across different trait categories. The red dotted line represents FDR-adjusted *P* = 0.05. Sample sizes vary for each trait. Boxplots provide, in order of bottom to top, the lower extreme, lower quartile, median, upper quartile, and upper extreme. **b** Forest plot of GTA association estimates (shown as points) and 95% FDR-adjusted two-sided Wald-type confidence intervals for six neonatal traits in ELGAN for nine genes with multiple TWAS GTAs across categories. The red line shows a null effect size of 0, and associations are colored blue for associations at FDR-adjusted *P* < 0.05. Sample size of 729 individuals from ELGAN were used to derived these estimates. **c** Follow-up GBAT and Mendelian randomization (MR) analysis results using RICHS data. On the left, effect size (shown as points) and 95% adjusted confidence intervals from GBAT (*x*-axis; two-sided *t*-test, df = 148) between GReX of RP-encoding genes and TWAS gene associations (pairs given on *y*-axis). On the right, MR effect size and 95% adjusted confidence interval (*x*-axis; two-sided Wald-type test) of RP-gene on TWAS gene (pairs on *y*-axis). The red line shows a null effect size of 0, and associations are colored blue for associations at FDR-adjusted *p* < 0.05. Sample size of 149 individuals from RICHS were used to derive these estimates.

detected negative associations between MDI and GReX of *RPL6* (−2.636, [−4.251, −1.02]) and *ERP29* (−3.332, [−4.987, −1.677]). As many of these genes encode for proteins involved in core processes (i.e., *RPL6* is involved in *trans*-activation of transcription and translation, and *SEC11A* has roles in cell migration and invasion)[45,46], understanding how the placental GReX of these genes affects neonatal traits may elucidate the potential long-lasting impacts of placental dysregulation.

**Body size and metabolic placental GTAs show trait associations in mice.** To further study functional consequences for selected TWAS-identified genes, we evaluated the 109 metabolic trait-associated genes in the Hybrid Mouse Diversity Panel (HMDP) for correlations with obesity-related traits[47]. This panel includes 100 inbred mice strains with extensive collection of obesity-related phenotypes from over 12,000 genes, with expression measured in a variety of adult tissues. Of the 109 genes, 73

were present in the panel and 36 showed significant cis-GReX associations with at least one obesity-related trait at FDR-adjusted $P < 0.10$ (Supplementary Data 11). For example, *EPB41L1* (*Epb4.1l1* in mice), a gene that mediates interactions in the erythrocyte plasma membrane, was associated with cholesterol and triglycerides in TWAS and showed 22 GReX associations with cholesterol, triglycerides, and HDL in mouse liver, adipose, and heart, with $R^2$ ranging between 0.09 and 0.31. Similarly, *UBC* (*Ubc* in mice), a ubiquitin maintaining gene, was associated with waist-hip ratio in the placental TWAS and showed 27 GReX associations with glucose in adults, insulin, and cholesterol in mouse aorta, liver, and adipose tissues in HMDP, with $R^2$ ranging between 0.08 and 0.14. Though generalizing these functional results from non-placental tissue in mice to humans is tenuous, we believe these 36 individually significant genes in the HMDP are fruitful targets for follow-up studies.

**MOSTWAS reveals functional hypotheses for distal placental regulation of GTAs.** An advantage of MOSTWAS's methodology is in functional hypothesis generation by identifying potential mediators that affect TWAS-identified genes. Using the distal-SNPs added-last test from MOSTWAS[22], we interrogated distal loci incorporated into expression models for trait associations, beyond the association at the local locus. For 88 of 248 associations, predicted expression from distal-SNPs showed significant associations at FDR-adjusted $P < 0.05$ (Fig. 3a [Step 4], Supplementary Data 6). For each significant distal association, we identified a set of biomarkers that potentially affects transcription of the TWAS gene: a total of 9 regulatory protein-encoding genes (RPs) and 159 CpG sites across all 89 distal associations. Particularly, we detected two RPs, *DAB2* (distal mediator for *PAPPA* and diastolic blood pressure, distal $Z = -3.98$) and *EPS15*, both highly expressed in placenta[48,49]. Mediated through *EPS15* (overall distal $Z = 7.11$ and 6.33, respectively), distally predicted expression of *SPATA13* and *FAM214A* showed association with waist-hip ratio. *EPS15* itself showed a TWAS association for waist-hip ratio (Supplementary Data 6), and the direction of the *EPS15* GTA was opposite to those of *SPATA13* and *FAM214A*. Furthermore, *RORA*, a gene encoding a transcription factor involved in inflammatory signaling[50], showed a negative association with transcription of *UBA3*, a TWAS gene for fetal birthweight. Low placental *RORA* expression was previously shown to be associated with lower birthweight[51]. Aside from functions related to transcription regulation, the 9 RPs (*CUL5, DAB2, ELL, EPS15, RORA, SLC2A4RG, SMARCC1, NFKBIA, ZC3H15*) detected by MOSTWAS were enriched for several ontologies (Supplementary Data 14), namely catabolic and metabolic processes, response to lipids, and multiple nucleic acid-binding processes[29].

As we observed strong correlations between expressions of RP-TWAS gene pairs in ELGAN (Supplemental Fig. S14), we then examined the associations between TWAS-identified genes and the GReX of any predicted mediating RPs in an external dataset. Using RICHS, we conducted a gene-based *trans*-eQTL scan using Liu et al.'s Gene-Based Association Testing (GBAT) method[52] to computationally validate RP-TWAS gene associations. We predicted GReX of the RPs using *cis*-variants through leave-one-out cross-validation and scanned for associations with the respective TWAS genes (Fig. 4c and Supplementary Data 15). We found a significant association between predicted *EPS15* and *FAM214A* expressions (effect size $-0.24$, FDR-adjusted $P = 0.019$). In addition, we detected a significant association between predicted *NFKBIA* and *HNRNPU* (effect size $-0.26$, FDR-adjusted $P = 1.9 \times 10^{-4}$). We also considered an Egger regression-based Mendelian randomization framework[53] in RICHS to estimate the

causal effects of RPs on the associated TWAS genes (Methods and Materials) using, as instrumental variables, *cis*-SNPs correlated to the RP and uncorrelated with the TWAS genes. We estimated significant causal effects for two RP-TWAS gene pairs (Fig. 5c and Supplementary Data 16): *EPS15* on *FAM214A* (causal effect estimate $-0.58$; 95% CI [0.21, 0.94]) and *RORA* on *UBA3* (0.58; [0.20, 0.96]). These GBAT and MR estimates between *EPS15* and *FAM214A* are in opposite directions of the simple correlations presented in Supplemental Fig. S14. However, as discussed in previous TWAS and MR studies[17,53], correlations between GReX and a phenotype are not equivalent to correlations between full expression and the phenotype, as full expression is subject multiple post-transcriptional process, while GReX is not.

We also examined the CpG methylation sites MOSTWAS marked as potential mediators for expression of TWAS genes for overlap with *cis*-regulatory elements in the placenta from the ENCODE Project Phase II[10], identifying 34 CpG sites (mediating 29 distinct TWAS genes) that fall in *cis*-regulatory regions (Supplementary Data 17). Interestingly, one CpG site mediating (cg15733049, Chromosome 1:2334974) *FAM214A* is found in low-DNase activity sites in placenta samples taken at various timepoints; additionally, cg15733049 is local to *EPS15*, the RP predicted to mediate genetic regulation of *FAM214A*. Furthermore, expression of *LARS2*, a TWAS gene for adult BMI, is mediated by cg04097236 (found within *ELOVL2*), a CpG site found in low DNase or high H3K27 activity regions; *LARS2* houses multiple GWAS risk SNPs for type 2 diabetes[54] and has shown adult BMI TWAS associations in other tissues[17,28]. Results from these external datasets add more evidence that these mediators play a role in gene regulation of these TWAS-identified genes and should be investigated experimentally in future studies.

**In vitro assays reveal widespread transcriptomic consequences of EPS15 knockdown.** Based on our computational results, we experimentally studied whether the inverse relationship between RP *EPS15* and its two prioritized target TWAS genes, *SPATA13* and *FAM214A*, is supported in vitro. We used a FANA oligonucleotide targeting *EPS15* to knock down *EPS15* expression in human placenta-derived JEG-3 choriocarcinoma cells and assessed the gene expression of the targets in no-addition controls, scramble oligo controls, and the knockdown variant via qRT-PCR. JEG-3 cells were selected for study based on their know first trimester-like phenotypes, including the synthesis and secretion of hCG, human placenta lactogen, progesterone, estrone, and estradiol[55,56]. Addition of FANA-EPS15 to JEG-3 cells decreased *EPS15* gene expression, while increasing the expression of *SPATA13* and *FAM214A* (50% decrease in *EPS15* expression, 795% and 377% increase in *SPATA13* and *FAM214A* expression, respectively). At FDR-adjusted $P < 0.10$, changes in gene expression of *EPS15* and downstream targets from the scramble were statistically significant against the knockdown oligo. Similarly, changes in gene expression between the control mRNA and RP and target mRNA were statistically significant (Fig. 6a).

To further investigate the transcriptomic consequences of *EPS15* knockdown in vitro, we measured transcriptome-wide gene expression in the choriocarcinoma cell lines via RNA-seq and conducted differential gene expression analysis across the knockdown cells and scramble oligo controls[57–59]. Owing to small sample sizes, we define a differentially expression gene with absolute $\log_2$-fold-change greater than 0.5 at $P < 1.32 \times 10^{-6}$, a Bonferroni correction across all assayed genes (Methods). We detected 650 genes downregulated and 838 genes upregulated in the *EPS15* knockdown cells, validating the negative correlations between *EPS15* and *SPATA13* and *FAM214A* observed in qRT-PCR (Fig. 6b and Supplementary Data 18–19). In particular, these

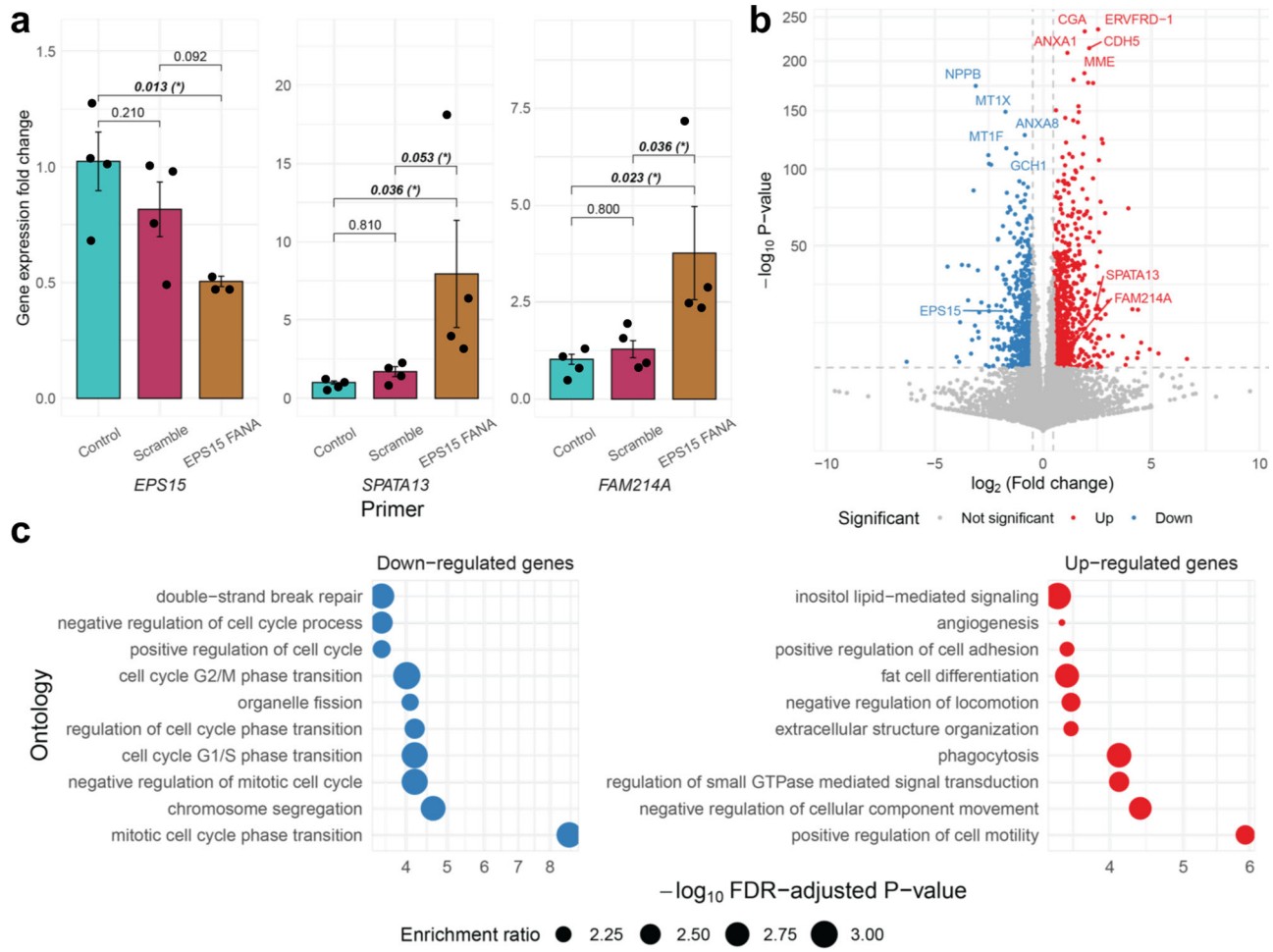

**Fig. 6 In vitro experiments in EPS15-knockdown human placenta-derived choriocarcinoma epithelial cells. a** Bar graph of the mean gene expression fold-changes with error bars of 1 standard deviation from the qRT-PCR from JEG-3 RNA. Nominal *P*-values of pairwise *t*-tests are shown, with an asterisk if Benjamini–Hochberg *FDR*-adjusted *P*<0.10 (*n* = 2 cells examined over two independent experiments). Note differences in *y*-axis scales. **b** Volcano plot of log₂ fold-change (*x*-axis) of differential expression across EP15 knockdown cells and scramble oligonucleotide against –log₁₀ FDR-adjusted *P*-value (*y*-axis; two-sided Wald-type tests from negative binomial regression). Upregulated genes are in red and downregulated genes in blue. Top up- and downregulated genes by *P*-value are labeled, as well as *EPS15, SPATA13*, and *FAM214A*. **c** Enrichment plot of over-representation of down- (blue) and upregulated (red) genes of PANTHER and KEGG pathways (*y*-axis) with –log₁₀ FDR-adjusted *P*-value (*x*-axis; one-sided Fisher's exact test). The size of the point gives the enrichment ratio.

downregulated genes were enriched for cell cycle, cell prolifera-tion, or replication ontologies, while upregulated genes were enriched for multiple different pathways, including lipid-related processes, cell movement, and extracellular organization (Fig. 5c and Supplementary Data 20–S21). Enrichments for cellular, molecular, and disease pathway ontologies support these enrich-ments (Supplemental Fig. S15 and Supplementary Data 20–S21). Though we could not study the effects of these three genes on body size-related traits, *cis*-GReX correlation analysis from the HMDP did reveal a negative *cis*-GReX correlation ($r = -0.31$, FDR-adjusted $P = 0.07$) between *Eps15* (mouse analog of human gene *EPS15*) and free fatty acids in mouse liver (Supplementary Data 13). These results prioritize *EPS15* for further study in larger cell line or animal studies as a potential regulator for multiple downstream genes, perhaps for genes affecting cell proliferation and replication in the placenta, like *SPATA13*[60].

## Discussion

The placenta has been understudied in large multi-tissue con-sortia efforts that study tissue-specific regulatory mechanisms[9,10]

relevant to complex trait etiology. To address this gap, we sys-tematically categorized placental gene-trait associations relevant to the DOHaD hypothesis using MOSTWAS, a method for enriching TWAS with distal genetic variants[22]. We detected 176 genes (enriched for cell growth and immune pathways) with transcriptome-wide significant associations, with the majority of GTAs linked to metabolic and neonatal/childhood traits. Fur-thermore, we could only estimate significantly positive placental GReX-mediated heritability for four neonatal traits but not for later-in-life traits. Many of these TWAS-identified genes, espe-cially those with neonatal GTAs, showed multiple GTAs across trait categories (nine genes with three or more GTAs). We examined phenome-wide GTAs for these nine genes in UKBB and found enrichments for traits affecting in immune and cir-culatory system (e.g., immune cell, erythrocyte, and platelet counts). We followed up with selected early-life traits in ELGAN and found associations with neonatal body size and infant cog-nitive development. These results suggest that placental expres-sion, mediated by fetal genetics, is most likely to have large effects on early-life traits, but these effects may persist later-in-life as etiologic antecedents for complex traits.

MOSTWAS also generates hypotheses for regulation of TWAS-detected genes, through distal mediating biomarkers, like transcription factors, miRNAs, or products downstream of CpG methylation islands[22]. Our computational results prioritized 89 GTAs with strong distal associations. We interrogated one such functional hypothesis: *EPS15*, a predicted RP-encoding gene in the EGFR pathway, regulates two TWAS genes positively associated with waist-hip ratio— *FAM214A*, a gene of unknown function, and *SPATA13*, a gene that regulates cell migration and adhesion[60–62]. In fact, *EPS15* itself showed a negative TWAS association with waist-hip ratio. In particular, *EPS15*, mainly involved in endocytosis, is a maternally imprinted gene and predicted to promote offspring health[49,63–65]. There is ample literature that implicates the protein product of *EPS15* as a direct or indirect transcription regulator. The protein Eps15 is an adaptor protein that regulates intracellular trafficking and has been detected in the nucleus of mammalian cells[66]. Once in the nucleus, Eps15 has shown to positively modulate transcription in a GAL4 transactivation assay[67]. Furthermore, Eps15 and its binding partner intersectin activate the Elk-1 transcription factor, pointing to Eps15's function in regulating gene expression in the nucleus[68]. Specific to the placenta, it has been proposed, through mouse models, that Eps15's interactions with multiple proteins suggest a role in cell adhesion of trophoblast to endothelial cells through biogenesis of exosomes and extracellular vesicles, a critical part of placental and fetal development[69–71].

In placental-derived choriocarcinoma epithelial cells, knockdown of *EPS15* showed increased expression of both *FAM214A* and *SPATA13*, as well as multiple genes involved in metabolic and hormone-related pathways. Though not implicating a direct causal effect, *EPS15*'s inverse association with *SPATA13* and *FAM214A* could provide more context to its full influence in placental developmental programming, perhaps by affecting cell proliferation or adhesion pathways. In vivo animal experiments, albeit limited in scope and generalizability, can be employed to further investigate GTAs, building off results from the HMDP showing cis-GReX correlations between *EPS15* mouse analog and fatty acid levels. Although these cis-GReX correlations from HMDP cannot be generalized from mice to humans, our in vitro assay provides valuable evidence for *EPS15* genomic regulation in the placenta. JEG-3 cells are reliable in use and provide accurate results when investigating specific cellular responses, such as the placental gene expression experiments used in this study; however, these cell lines do not capture interactions between cell types in the placental tissue and its effects on the placental transcriptome, as a whole. Our results also support the potential of MOSTWAS to build mechanistic hypotheses for upstream regulation of TWAS genes that hold up to experimental rigor.

We conclude with limitations of this study and future directions. First, our analysis considers only placental tissue. Though many of our GTAs leverage distal-eQTL architecture, which tend to be tissue-specific, the QTLs we leverage in TWAS may not be placenta-specific. A similar analysis across developmental and adult tissues could reveal more widespread genetic signals associated with these traits. Second, the ELGAN Study gathered molecular data from infants born extremely pre-term. If unmeasured confounders affect both prematurity and a trait of interest, GTAs could be subject to backdoor collider confounding[72]. However, significant TWAS genes did not show associations for gestational duration, suggesting minimal bias from this collider effect. An extensive comparison of genome-wide eQTL architecture between ELGAN and RICHS, highlighting differences in genetic effects on gene expression across pre-term status, could be of particular scientific importance. An interesting future endeavor could include negative control variables to account for unmeasured confounders in predictive models to allow for more generalizability

of predictive models[73,74]. Fourth, though we did scan neonatal traits in ELGAN using individual-level genotypes, as the sample size is small, larger GWAS with longitudinal traits could allow for rigorous Mendelian Randomization studies that investigate relationships between traits across the life course, in the context of placental regulation. Fifth, we curated a list of regulatory proteins to include as potential mediators but use RNA expression of the genes that code for these proteins as a proxy for abundance. We contend that RNA abundance of the gene is a noisy estimate of the protein abundance. An interesting extension of this analysis could consider a proteome-wide association study, using the MOSTWAS framework to identify protein interactions that are disease-related. Lastly, due to small sample sizes of other ancestry groups in ELGAN, we could only credibly impute expression into samples from European ancestry and our TWAS only considers GWAS in populations of European ancestry[75]. We emphasize acquisition of larger genetic and genomic datasets from understudied and underserved populations, especially related to early-in-life traits.

Our findings reveal functional evidence for the fundamental influence of placental genetic and genomic regulation on developmental programming of early- and later-in-life traits, identifying placental gene-trait associations and testable functional hypotheses for upstream placental regulation of these genes. Future large-scale tissue-wide studies should emphasize the placenta as a core tissue for learning about the developmental origins of health and disease.

## Methods

The study was approved by the Institutional Review Board of the University of North Carolina at Chapel Hill (IRB #16-2535). All participants consented to the study as per IRB protocol.

### Data acquisition and quality control

*Genotype data.* Genomic DNA was isolated from umbilical cord blood and genotyping was performed using Illumina 1 Million Quad and Human OmniExpression-12 v1.0 arrays[34,76]. Prior to imputation, from the original set of 731,442 markers, we removed SNPs with call rate <90% and MAF < 1%. We only consider genetic variants on autosomes. We did not use deviation from Hardy–Weinberg equilibrium as an exclusion criterion since ELGAN is an admixed population. This resulted in 700,845 SNPs. We removed 4 individuals out of 733 with sample-level missingness >10% using PLINK[77]. We first performed strand-flipping according to the TOPMed Freeze 5 reference panel and using eagle and minimac4 for phasing and imputation[78–80]. Genotypes were coded as dosages, representing 0, 1, and 2 copies of the minor allele. The minor allele was coded in accordance with the NCBI Database of Genetic Variation[81]. Overall, after QC and normalization, we considered a total of 6,567,190 SNPs. We obtained processed genetic data from the Rhode Island Children's Health Study, as described before[36].

*Expression data.* mRNA expression was determined using the Illumina QuantSeq 3' mRNA-Seq Library Prep Kit, a method with high strand specificity[82]. mRNA-sequencing libraries were pooled and sequenced (single-end 50 bp) on one lane of the Illumina HiSeq 2500. mRNA were quantified through pseudo-alignment with salmon[57] mapped to the GENCODE Release 31 (GRCh37) reference transcriptome. miRNA expression profiles were assessed using the HTG EdgeSeq miRNA Whole Transcriptome Assay (HTG Molecular Diagnostics, Tucson, AZ). miRNA were aligned to probe sequences and quantified using the HTG EdgeSeq System[83].

Genes and miRNAs with <5 counts for each sample were filtered, resulting in 12,020 genes and 2047 miRNAs for downstream analysis. We only consider autosomal genes and miRNAs. Distributional differences between lanes were first upper-quartile normalized[84,85]. Unwanted technical and biological variation (e.g., tissue heterogeneity) was then estimated using RUVSeq[86], where we empirically defined transcripts not associated with outcomes of interest as negative control housekeeping probes[87]. One dimension of unwanted variation was removed from the variance-stabilized transformation of the gene expression data using the limma package[59,86–88]. We obtained pre-processed RNA expression data from the Rhode Island Children's Health Study, as described before[36]. Pre-processing steps for RNA expression data from the RICHS are different from those employed here in the ELGAN study (e.g., using EDASeq and edgeR for GC bias correction and normalization[35]); differences in pre-processing may affect inferred distal-eQTL architecture, as cell-type heterogeneity is captured and removed differently across ELGAN and RICHS[22,89,90].

*Methylation data.* Extracted DNA sequences were bisulfate-converted using the EZ DNA methylation kit (Zymo Research, Irvine, CA) and followed by quantification using the Infinium MethylationEPIC BeadChip (Illumina, San Diego, CA), which measures CpG loci at a single nucleotide resolution, as previously described[91–94]. Quality control and normalization were performed resulting in 856,832 CpG probes from downstream analysis, with methylation represented as the average methylation level at a single CpG site (β-value)[92,95–98]. DNA methylation data was imported into R for pre-processing using the minfi package[96,97]. Quality control was performed at the sample level, excluding samples that failed and technical duplicates; 411 samples were retained for subsequent analyses.

Functional normalization was performed with a preliminary step of normal-exponential out-of band (noob) correction method[99] for background subtraction and dye normalization, followed by the typical functional normalization method with the top two principal components of the control matrix[96,97]. Quality control was performed on individual probes by computing a detection *P*-value and excluded 806 (0.09%) probes with non-significant detection ($P > 0.01$) for 5% or more of the samples. A total of 856,832 CpG sites were included in the final analyses. Lastly, the ComBat function was used from the sva package to adjust for batch effects from sample plate[100]. In addition, to account for cell-type heterogeneity, 5 surrogate values were estimated and removed from the data to account using the sva package, as previously described[15,92,100]. The data were visualized using density distributions at all processing steps. Each probe measured the average methylation level at a single CpG site. Methylation levels were calculated and expressed as β values, with

$$\beta = \frac{M}{U + M + 100} \quad (1)$$

where $M$ is the intensity of the methylated allele and $U$ is the intensity of the unmethylated allele. β-values were logit transformed to $M$ values for statistical analyses[101]. Overall, after QC and normalization, we considered 846,233 CpG sites, only on autosomes.

*Differences in inclusion/exclusion criteria between ELGAN and RICHS.* We highlight some differences in inclusion and exclusion criteria employed by ELGAN and RICHS. ELGAN enrolled children born extremely pre-term (<28 weeks gestation) and surviving 28 days postnatally, with full details of the study recruitment and descriptive statistics of the cohort in O'Shea et al.[21]. In contrast, as mentioned in Peng et al.[36], the RICHS sample consists of term infants (≥37 weeks gestation, not twins) born without serious pregnancy complications or congenital and chromosomal abnormalities. In addition, RICHS oversampled for large-for-gestational age (LGA, >90% 2013 Fenton Growth Curve) and small-for-gestational age (SGA, <10% 2013 Fenton Growth Curve) infants.

*GWAS summary statistics.* Summary statistics were downloaded from the following consortia: the UK Biobank[23], Early Growth Genetics Consortium[24], Genetic Investigation of Anthropometric Traits[25], Psychiatric Genomics Consortium[26], and the Complex Trait Genetics Lab[27] (Supplementary Data 1). Genomic coordinates were transformed to the hg38 reference genome using liftOver[102,103]. SNP heritability for each trait and genetic correlations for all pairwise combinations of traits were estimated using LD-score regression with the European ancestry sample from the 1000 Genomes Project as a reference for LD scores[31,32].

*QTL mapping.* The first step in the MOSTWAS pipeline is to scan for associations between SNPs and genes (genome-wide eQTL analysis) and between mediators and genes. We conducted genome-wide eQTL mapping between all genotypes and all genes in the transcriptome using a standard linear regression in MatrixQTL[104]. Here, we ran an additive model with gene expression as the outcome, SNP dosage as the primary predictor of interest, with covariate adjustments for 20 genotype PCs (for population stratification), sex, gestational duration, maternal age, maternal smoking status, and 10 expression PEER factors[105]. Mediators here are defined as RNA expression of genes that code for regulatory proteins (curated in TFcheckpoint[33]), miRNAs, and monomorphic CpG methylation sites. In sum, we call the expression or methylation of a mediator its intensity. We also conducted genome-wide mediator-QTL mapping with the intensity of mediators as the outcome with the same predictors as in the eQTL mapping. Lastly, we also assessed associations between mediators and gene expression using the same linear models, with mediator intensity as the main predictor. All intensities were scaled to zero mean and unit variance.

**Estimation of SNP heritability of gene expression.** An important step in a TWAS pipeline is estimation of SNP heritability of expression, as SNP heritability is a strong determinant of TWAS study power[17,106]. Heritability using genotypes within 1 Megabase of the gene of interest and any prioritized distal loci was estimated using the GREML-LDMS method, proposed to estimate heritability by correction for bias in LD in estimated SNP-based heritability[107]. Analysis was conducted using GCTA v1.93.1[108]. Briefly, Yang et al. shows that estimates of heritability are often biased if causal variants have a different minor allele frequency (MAF) spectrums or LD structures from variants used in analysis. They proposed an LD and MAF-stratified GREML analysis, where variants are stratified into groups by MAF and LD, and genetic relationship matrices (GRMs) from these variants in each group are jointly fit in a multi-component GREML analysis.

**Gene expression models.** We used MOSTWAS to train predictive models of gene expression from germline genetics, including distal variants that were either close to associated mediators (transcription factors, miRNAs, CpG sites) or had large indirect effects on gene expression[22] (Supplemental Fig. S1). Our assumption here is that distal-eQTLs of a gene that are local to transcription factor-encoding genes, miRNAs, or regulatory features local to CpG methylation sites may be potentially mediated by cis-QTLs to these local features. This assumption has been employed by multiple studies previously to identify trans-eQTLs in multiple tissues[89,109–111]. For CpG methylation sites, we used the maxprobes R package to filter out cross-reactive or polymorphic probes, which may induce bias[112–114]. MOSTWAS contains two methods of predicting expression: (1) mediator-enriched TWAS (MeTWAS) and (2) distal-eQTL prioritization via mediation analysis. For MeTWAS, we first identified mediators strongly associated with genes through correlation analyses between all genes of interest and a set of distal mediators (FDR-adjusted $P<0.05$). We then trained local predictive models (using SNPs within 1 Mb) of each mediator using either elastic net or linear mixed model, used these models to impute the mediator in the training sample, and included the imputed values for mediators as fixed effects in a regularized regression of the gene of interest. For DePMA, we first conducted distal-eQTL analysis to identify all distal-eQTLs at $P<10^{-6}$ and then local mediator-QTL analysis to identify all mediator-QTLs for these distal-eQTLs at FDR-adjusted $P<0.05$. We tested each distal-eQTL for their absolute total mediation effect on the gene of interest through a permutation test and included eQTLs with significantly large effects in the final expression model. Full mathematical details are provided in Bhattacharya et al.[22]. We considered only genes with significantly positive heritability at nominal $P<0.05$ using a likelihood ratio test and fivefold McNemar's adjusted cross-validation $R^2 \geq 0.01$, a cross-validation cutoff used by many previous TWAS analyses[16,17,28,36,75,115,116]. McNemar's adjustment to the traditional $R^2$ is computed as

$$R_{adjusted}^2 = 1 - \left(1 - R^2\right)\frac{n - 1}{n - \nu - 1} \quad (2)$$

where $n$ is the sample size and $\nu$ is the number of predictors in this linear model. Since this $R^2$ is computed only between the observed and predicted expression values, $\nu = 1$.

**TWAS tests of association**

*Overall TWAS test.* In an external GWAS panel, if individual SNPs are available, model weights from MeTWAS or DePMA can be multiplied by their corresponding SNP dosages to construct the Genetically Regulated eXpression (GReX) for a given gene. This value represents the portion of expression (in the given tissue) that is directly predicted or regulated by germline genetics. We run a linear model or test of association with phenotype using this GReX value for the eventual TWAS test of association.

If individual SNPs are not available, then the weighted burden Z-test, proposed by Gusev et al., can be employed using summary statistics[17]. Briefly, we compute

$$\widetilde{Z} = \frac{w_G Z}{\left(w_G \Sigma_{s,s} w_G^T\right)^{1/2}} \quad (3)$$

Here, $Z$ is the vector of $Z$-scores of SNP-trait associations for SNPs used in predicting expression. The vector $w_G$ represents the vector of SNP-gene effects from MeTWAS or DePMA and $\Sigma_{s,s}$ is the LD matrix (correlation matrix between genotypes) between the SNPs represented in $w_G$. The test statistic $\widetilde{Z}$ can be compared to the standard Normal distribution for inference.

*Permutation test.* We implement a permutation test, condition on the GWAS effect sizes, to assess whether the same distribution of SNP-gene effect sizes could yield a significant associations by chance[17]. We permute $w_G$ 1000 times without replacement and recompute the weighted burden test to generate a null distribution for $\widetilde{Z}$. This permutation test is only conducted for overall associations at $P<2.5 \times 10^{-6}$.

*Distal-SNPs added-last test.* Lastly, we also implement a test to assess the information added from distal-eSNPs in the weighted burden test beyond what we find from local SNPs. This test is analogous to a group added-last test in regression analysis, applied here to GWAS summary statistics. Let $Z_l$ and $Z_d$ be the vector of $Z$-scores from GWAS summary statistics from local and distal-SNPs identified by a MOSTWAS model. The local and distal-SNP effects from the MOSTWAS model are represented in $w_l$ and $w_d$. Formally, we test whether the weighted $Z$-score $\widetilde{Z}_d = w_d Z_d$ from distal-SNPs is significantly larger than 0 given the observed weighted $Z$-score from local SNPs $\widetilde{Z}_l = w_l Z_l$. We draw from the assumption that $(\widetilde{Z}_d, \widetilde{Z}_l)$ follow a bivariate Normal distribution. Namely, we conduct a two-sided Wald-type test for the null hypothesis:

$$H_0 : w_d Z_d | w_l Z_l = \widetilde{Z}_l = 0 \quad (4)$$

We can derive a null distribution using conditionals of bivariate Normal distributions; see Bhattacharya et al.[22].

**Genetic heritability and correlation estimation**. At the genome-wide genetic level, we estimated the heritability of and genetic correlation between traits via summary statistics using LD-score regression[31]. On the predicted expression level, we adopted approaches from Gusev et al.[17] and Mancuso et al.[28] to quantify the heritability ($h^2_{GE}$) of and genetic correlations ($\rho_{GE}$) between traits at the predicted placental expression level. We assume that the expected $\chi^2$ statistic under a complex trait is a linear function of the LD-score[31]. The effect size of the LD-score on the $\chi^2$ is proportional to $h^2_{GE}$:

$$E[\chi^2] = 1 + \left(\frac{N_T l}{M}\right) h^2_{GE} + N_T a \tag{5}$$

where $N_T$ is the GWAS sample size, $M$ is the number of genes, $l$ is the LD scores for genes, and $a$ is the effect of population structure. We estimated the LD scores of each gene by predicting expression in European samples of 1000 Genomes and computing the sample correlations and inferred $h^2_{GE}$ using ordinary least squares. We employed RHOGE to estimate and test for significant genetic correlations between traits at the predicted expression level[28].

**Multi-trait scans in UKBB and ELGAN**. For nine genes with three or more associations across traits of different categories, we conducted multi-trait TWAS scans in UK Biobank. Here, we used the weighted burden test in UKBB GWAS summary statistics from samples of European ancestry for 296 traits grouped by ICD code blocks (circulatory, congenital malformations, immune, mental disorders, musculoskeletal, neonatal, neurological, and respiratory). We also imputed expression for these genes in ELGAN using 729 samples with individual genotypes and conducted a multi-trait scan for 6 neonatal traits: neonatal chronic lung disease, head circumference Z-score, fetal growth restriction, birthweight Z-score, necrotizing enterocolitis, and Bayley II Mental Development Index (MDI) at 24 months. For continuous traits (head circumference Z-score, birthweight Z-score, and mental development index), we used a simple linear regression with the GReX of the gene as the main predictor, adjusting for race, sex, gestational duration (in days), inflammation of the chorion, and maternal age. For binary traits, we used a logistic regression with the same predictors and covariates. These covariates have been previously used in placental genomic studies of neonatal traits because of their strong correlations with the outcomes and with placental transcriptomics and methylomics[15,92,117].

**Validation analyses in RICHS**. Using genotype and RNA-seq expression data from RICHS[36], we attempted to validate RP-TWAS gene associations prioritized from the distal-SNPs added-last test in MOSTWAS. We first ran GBAT, a trans-eQTL mapping method from Liu et al.[52] to assess associations between the loci around RPs and the expression of TWAS genes in RICHS. GBAT tests the association between the predicted expression of a RP with the expression of a TWAS gene, improving power of trans-eQTL mapping[118]. We also conduct directional Egger regression-based Mendelian randomization to estimate and test the causal effects of the expression of the RP on the expression of the TWAS gene[119].

**Human Mouse Diversity Panel**. To provide some functional evidence of gene associations with metabolic traits, we evaluated the 109 metabolic trait-associated genes from our human placental TWAS in the Hybrid Mouse Diversity Panel (HMDP) for correlations with obesity-related traits in mice[47]. This panel includes 100 inbred mice strains with extensive collection of obesity-related phenotypes (e.g., cholesterol, body fat percentage, insulin, etc) from over 12,000 genes, with expression measured in a variety of adult tissues (liver, adipose, aorta). We note that the HMDP only considers adult tissues and does not include placental gene expression. In the HMDP, we consider both trait correlation to tissue-specific gene expression and cis-GReX (genetically regulated expression controlled by cis-eQTLs).

**In vitro functional assays**

*Cell culture and treatment.* The JEG-3 choriocarcinoma cells were purchased from the American Type Culture Collection (Manassas, VA; ATCC HTB-36). Cells were grown in Gibco RMPI 1640, supplemented with 10% fetal bovine serum (FBS), and 1% penicillin/streptomycin at 37 °C in 5% $CO_2$. Cells were plated at $2.1 \times 10^6$ cells per 75 cm³ flask and incubated under standard conditions until achieving roughly 90% confluence. To investigate the effects of gene silencing, we used AUMsilence FANA oligonucleotides for mRNA knockdown of *EPS15* (AUM Bio Tech, Philadelphia, PA) and subsequent analysis of predicted downstream target genes *SPATA13* and *FAM214A*. On the day of treatment, cells were seeded in a 24-well culture plate at $0.05 \times 10^6$ cells per well. Cells were plated in biological duplicate. FANA oligos were dissolved in nuclease-free water to a concentration of 500 μM, added to cell culture medium to reach a final concentration of 20 μM and incubated for 24 h at 37 °C in 5% $CO_2$.

*mRNA expression by quantitative real-time polymerase chain reaction and RNA Sequencing.* Treated and untreated JEG-3 cells were harvested in 350 μL of buffer RLT plus. Successive RNA extraction was performed using the AllPrep DNA/RNA/miRNA Universal Kit according to the manufacturer's protocol. RNA was

quantified using a NanoDrop 1000 spectrophotometer (Thermo Scientific, Waltham, MA). RNA was then converted to cDNA, the next step toward analyzing gene expression. Next, mRNA expression was measured for *EPS15*, *SPATA13*, and *FAM214A* using real-time qRT-PCR and previously validated primers. Samples were run in technical duplicate. Real-time qRT-PCR Ct values were normalized against the housekeeping gene B-actin (*ACTB*), and fold-changes in expression were calculated based on the ΔΔCT method[120]. Each sample was prepared in biological duplicate and technical duplicate. These samples were pooled together for sequencing to yield data representing four samples per exposure group. Fold-change calculations using the Delta Delta CT method was calculated for each sample individually:

$$\text{Delta CT}_{\text{treated}} = \text{CT}_{\text{GOI,treated}} - \text{CT}_{\text{House,treated}} \tag{6}$$

Treated and untreated samples of JEG-3 RNA previously extracted using the AllPrep DNA/RNA/miRNA Universal Kit were submitted to the High-Throughput Sequencing Facility at UNC Chapel Hill for RNA sequencing. Total RNA samples were submitted for sequencing using the HS4000 HO platform. Samples were sequenced in duplicate, and libraries were prepped with the Kapa Stranded mRNA-Seq kit from Illumina Platforms. Sequencing was performed after all samples passed QAQC, with a paired-end read type, with a read length of $2 \times 75$.

*Statistical analysis.* Statistical analysis was performed using a one-way ANOVA (with nominal significance level $\alpha = 0.05$). Post-hoc pairwise $t$-tests (three degrees of freedom for biological and technical duplicate) were utilized to investigate direct comparisons within sample groups.

*Differential expression analysis.* RNA-seq quantified counts (transcripts per kilobase million) were imported using tximeta[58] and summarized to the gene-level. Differential expression analysis between *EPS15* knockdown samples and scramble oligo controls was conducted using DESeq2[59]. Although false-positive rates are well-controlled even at low sample sizes[121], true-positive rates at such a low sample size are low for smaller thresholds of log-transformed fold-changes. Thus, guided by Schurch et al.'s analysis, due to very limited sample size, we considered a gene to be differentially expressed if the absolute log₂-fold-change is greater than 1 and $P < 0.05/37,788 = 1.32 \times 10^{-6}$. This $P$-value threshold is a strict Bonferroni threshold across 37,788 quantified genes.

**Reporting summary**. Further information on research design is available in the Nature Research Reporting Summary linked to this article.

## Data availability

ELGAN mRNA, miRNA, and CpG methylation data can be accessed from the NCBI Gene Expression Omnibus GSE154829 and GSE167885. ELGAN genotype data is protected, as subjects are still enrolled in the study; any inquiries or data requests must be made to RCF and HPS. Specifically generated for this study, RNA-seq data for placental JEG-3 cells are deposited in the NCBI GEO Database under accession code GSE185071. GWAS summary statistics can be accessed at the following links: UK Biobank (https://pan.ukbb.broadinstitute.org/downloads), GIANT consortium (https://portals.broadinstitute.org/collaboration/giant/index.php/GIANT_consortium_data_files), PGC (https://www.med.unc.edu/pgc/download-results/), EGG consortium (https://egg-consortium.org/), and CTG Lab (https://ctg.cncr.nl/documents/p1651/SavageJansen_IntMeta_sumstats.zip). The RICHS eQTL dataset can be accessed via dbGaP accession number phs001586.v1.p1. Placental epigenomic annotations from the ENCODE Project are available from https://www.encodeproject.org/, with specific accession numbers in Supplementary Data 17 (13 different accession numbers). All models and full TWAS results can be accessed at https://doi.org/10.5281/zenodo.4618036[122].

## Code availability

Sample scripts for analysis are provided at https://github.com/bhattacharya-a-bt/dohad_twas. The MOSTWAS software is accessible at https://bhattacharya-a-bt.github.io/MOSTWAS/articles/MOSTWAS_vignette.html. We use the following software in this study: PLINK v1.9 (https://zzz.bwh.harvard.edu/plink/), eagle v2.4.1 (https://alkesgroup.broadinstitute.org/Eagle/), minimac4 (https://genome.sph.umich.edu/wiki/Minimac4), Salmon 1.5.1 (https://salmon.readthedocs.io/en/latest/salmon.html), HTG EdgeSeq System (https://www.htgmolecular.com/systems/edgeseq), RUVSeq 3.12 (https://bioconductor.org/packages/release/bioc/html/RUVSeq.html), limma 3.12 (https://bioconductor.org/packages/release/bioc/html/limma.html), minfi 3.12 (https://bioconductor.org/packages/release/bioc/html/minfi.html), sva 3.40 (https://bioconductor.org/packages/release/bioc/html/sva.html), liftover 1.16.0 (https://www.bioconductor.org/packages/release/workflows/html/liftOver.html), ldsc 1.0.1 (https://github.com/bulik/ldsc), MOSTWAS 1.0.0 (https://github.com/bhattacharya-a-bt/MOSTWAS), MatrixEQTL 2.3 (https://cran.r-project.org/web/packages/MatrixEQTL/index.html), GCTA GREML-LDMS v1.93.1 (https://yanglab.westlake.edu.cn/software/gcta/#Overview), RHOGE 2018-02-28 (https://github.com/bogdanlab/RHOGE), maxprobes 0.0.1 (https://github.com/markgene/maxprobes), txmeta 3.14 (https://bioconductor.org/packages/release/bioc/html/tximeta.html), DESeq2 3.14 (https://bioconductor.org/packages/release/bioc/html/DESeq2.html), MendelianRandomization 0.5.1 (https://cran.r-project.org/web/packages/MendelianRandomization/index.html), and GBAT (https://github.com/xuanyao/GBAT).

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

## Acknowledgements

We thank Michael Love, Kanishka Patel, Michael Gandal, Chloe Yap, Bogdan Pasaniuc, and Jon Huang for their thoughts about the research. We also thank the following consortia and research groups for their publicly available GWAS summary statistics, eQTL datasets, and/or epigenomic annotations: the UK Biobank and the Neale Lab, the Genetic Investigation of Anthropometric Traits Consortium, the Psychiatric Genetics Consortium, the Early Growth Genetics Consortium, the Complex Trait Genetics Lab, the Rhode Island Child Health Study, and the ENCODE Project. This study was supported by grants from the National Institutes of Health (NIH), specifically the National Institute of Neurological Disorders and Stroke (U01NS040069; R01NS040069), the Office of the NIH Director (UG3OD023348), the National Institute of Environmental Health Sciences (T32-ES007018; P30ES019776; R24ES028597), the National Heart, Lung and Blood Institute (R01HL47883, R01HL148577), the National Institute of Nursing Research (K23NR017898; R01NR019245), and the Eunice Kennedy Shriver National Institute of Child Health and Human Development (R01HD092374; R03HD101413; P50HD103573).

## Author contributions

Conceptualization: A.B., T.M.O., R.C.F., H.P.S.; Data curation: A.B., A.N.F., V.A., W.L., Y.L., C.P., C.J.M., T.M.O., R.C.F., H.P.S.; Formal analysis: A.B., A.N.F., W.L., H.P.S.; Funding acquisition: A.J.L., Y.L., R.M.J., L.S., K.C.K.K., C.J.M., T.M.O., R.C.F., H.P.S.; Investigation: A.B., A.N.P., H.J.H., R.C.F., H.P.S.; Methodology: A.B., Y.L., H.P.S;. Project administration: A.B., R.C.F., H.P.S.; Resources: T.M.O., R.C.F., H.P.S.; Software: A.B., W.L., Y.L.; Supervision: A.B., Y.L., R.C.F., H.P.S.; Validation: A.B., C.J.M.; Visualization: A.B., A.N.F., R.H.; Writing—original draft: A.B., R.C.F., HPS; Writing—review & editing: A.B., A.J.L., A.N.F., V.A., R.H., W.L., Y.L., R.M.J., L.S., H.JH., K.C.K.K., C.J.M., T.M.O., R.C.F., H.P.S.

## Competing interests

The authors declare no competing interests.

## Additional information

**Peer review information** *Nature Communications* thanks Jingjing Yang, Ke Hao and the other anonymous reviewer for their contribution to the peer review this work. Peer reviewer reports are available.

