## [Peer Review File · Nature Communications]

REVIEWER COMMENTS

Reviewer #1 (Remarks to the Author):

This manuscript applied a recently proposed TWAS method to identify gene-trait associations that are potentially mediated through multi-omics of placenta, while accounting for distal eQTL. Significant gene-trait associations were identified, mostly for neonatal and metabolic traits. Follow-up studies show these significant gene-trait associations were enriched in cell growth and immunological pathways. The authors conducted comprehensive analyses to investigate these significant gene-trait associations. Overall, the findings are profound. But it is not clear what is the cohesive key contribution of this paper to our understanding of these investigated traits with respect to placental multi-omics variations. Here are my major concerns:

1. It is not clear how multi-omics data besides mRNA expression, like CpG methylation sites and miRNA expression data, were considered by MOSTWAS method.
2. The number of ~2,994 significant gene models seems quite a small proportion comparing to the total ~20K genes. Would a different method for training the gene expression prediction model be able to provide a greater number of significant gene models, like TIGAR (PMCID: PMC6698804), PMR (<https://www.nature.com/articles/s41467-020-17668-6>).
3. Please state clearly which tool/method was used to estimate heritability in this manuscript and plot the 95% confidence intervals of the heritability estimates.
4. A heatmap with all pair-wise correlation test results would be better than the current Figure 3C plot.
5. A phewas-like Manhattan plot with respect to each of the examined genes would provide a more complete view of the analysis results for the genome-wide association study in early- and later-life traits.
6. Please state if the LD matrix $\Sigma(S,S)$ is the genotype correlation matrix or covariance matrix. Actually, for using the Z-score statistic as mentioned in the Methods for TWAS, the genotype correlation matrix should be used.

Reviewer #3 (Remarks to the Author):

The overall goal of this paper was to characterize placental gene-trait associations relevant to the DOHAD hypothesis using the newly developed MOSTWAS method for eQTL analysis. The major strengths of this paper are the robust transcriptomics data and efforts to integrate this data using novel computational approaches. This paper is significant as it provides evidence of changes in placental gene expression related to genetic polymorphisms associated with metabolic and childhood/neonatal traits, supporting the DOHAD hypotheses. Overall, this paper is very ambitious and tackles many aspects of genetic control of placental gene expression, but is challenging to follow due to the sheer number of analyses being performed (RHOGE, MOSTWAS, FOCUS, GBAT, Mouse Diversity Panel, GREML-LDMS, MatrixQTL...). Overall, It would be helpful if the authors could go through and further clarify the goals of each analysis. There are some additional areas that could be further explained in this paper, which are highlighted in the following comments:

1. Demographic data on the ELGAN cohort is presented in Supplemental Table S4, but there is no demographic data provided on the RICHs cohort, which was used to validate the findings of gene expression prediction. How do the RICHs and ELGAN cohorts compare in terms of race, ethnicity, SES, rates of premature birth, and other key perinatal outcomes? It would be helpful to mention in the discussion about the differences in these cohorts and the validity of the RICHs cohort as an independent validation cohort.
2. Did the RNA sequencing data from the RICHs cohort undergo the same preprocessing steps as

the ELGAN cohort? (such as RUV-seq) This information is not described in the methods.

3. How did the authors handle cross-reactive/polymorphic probes in their analysis? These probes introduce bias in studies utilizing Illumina EPIC arrays (doi: <https://doi.org/10.1093/nargab/lqaa105>), and are frequently removed prior to downstream analyses.

4. How were the X and Y chromosome handled in this analysis?

5. How did the authors define transcription factors for MOSTWAS?

6. The authors address cellular heterogeneity present in placental omics data by adjusting for unwanted biological variation using RUVSeq. However, the authors do not adjust for this same variability in the DNA methylation data, and there are established methods to do this in methylation data (Houseman et al, 2015, Yuan et al, 2020)

7. For the heritable traits, can you provide some clarity if the GWAS results (HDL, Glucose, Cholesterol) that directly related to blood markers/metabolites were quantified in adults or children in the studies which they were derived? Also, the way this is presented may make it seem to a casual reader that these measurements were collected in the same population based on the labeling in some of the tables/figures (such as supplemental figure S9) and some of the wording (For example page 8 line 27-page 9 line 1). Readability could be improved by slight rewording.

How was the threshold of adjusted $R^2 > 0.01$ selected? These values of the cross validation and out of sample R^2 seem low-is there an example of another paper that uses this threshold? Also, in Page 5 Line 25, the authors refer to an "adjusted" R^2 -How was this adjusted?

8. The authors perform in vitro validation using JEG-3 Cells, which are not immortalized trophoblast cell lines. The JEG-3 cells are derived from a choriocarcinoma cell line that is used to model the placental trophoblast because they synthesize placental hormones and express key placental hormones including HLA-G. There are other cell lines, such as SWAN-71 and HTR8-SV/neo cells, which are derived from immortalized trophoblast cells. Thus, this cell line should be referred to appropriately in the manuscript text. What was the justification for using this particular cell line?

9. Can the authors comment more on the logic of selecting the gene EPS15 for experimental validation, out of the 9 TFs they identified? The authors state that this gene is highly expressed in the placenta, but the citation they reference (#45 <https://doi.org/10.1093/humupd/dmq052>) only mentions EPS15 once in a table of imprinted genes, using data derived from an imprinted gene catalogue (www.geneimprint.com). The fact that it is imprinted raises additional questions about how this is incorporated in the results. From what I can discern, EPS15 is a substrate for the EGFR receptor, and has been shown to localize within the nucleus (doi:10.1186/1478-811X-7-24), but I cannot find any literature directly implicating it as a transcription factor. Moreover, it is not included in commonly utilized database of transcription factors (<https://doi.org/10.1016/j.cell.2018.01.029> , <https://doi.org/10.1038/s41588-019-0411-1>). More insight into the function of this gene in the placenta and its role in transcriptional regulation (overall) is warranted, as it was highlighted as one of the main findings in the abstract.

10. The authors describe experiments where EPS15 knockdown causes increased expression of genes SPATA13 and FAM214A (Page 13, Line 2, and in Figure 5B). This would indicate that this gene is a negative regulator, since less activity of this TF is leading to increased expression. However, in supplemental figure 13, they show positive correlations between EPS15 and SPATA13 and FAM214A in the RICHS cohort, which would implicate it as a positive regulator (Increased TF expression, Increased expression of downstream genes) So the data they are presenting here appears contradictory. Can the authors explain this?

11. In the methods, the authors describe confirming gene expression changes in JEG-3 cells through RT-PCR (Page 23, Lines 4-5). However, it appears later in the methods (Page 23, Lines 13-16) that differential gene expression was performed, which is presented in the results and in

Figure 5B. However, in the methods there isn't any information provided about how RNA was isolated from the JEG-3 cells, or any details about the sequencing (i.e library prep method, sequencer, QC). Please provide this information.

12. As a follow up to this, the authors plated cells in duplicate, then performed siRNA isolation in duplicate. For the real time PCR, the samples were also run in technical duplicate (Page 23, line 2). The authors then had 3 degrees of freedom for their biological and technical duplicates (Page 23 Line 10) for pairwise T tests. Can the authors describe in more detail how the technical and biological duplicates were used to calculate the fold change using the Delta Delta CT method. Was this calculated separately for each sample? Also, if the authors also did RNA sequencing, was this performed only in duplicate? Normally the minimum number of samples presented in RNA sequencing experiments with in vitro samples are triplicates. (Example: <https://doi.org/10.1111/aji.12722>)

13. The authors include an analysis using the Hybrid Mouse diversity panel to look at correlations with obesity related traits. However, the authors should mention in the discussion the limitations of generalizing between mouse and human studies of the placenta, given that the biology is so different.

14. What do the authors mean when they say "collection of negative control variables"-page 15, line 12.

15. Moreover, in the discussion, the authors should discuss the assumptions made in this analysis, particularly the assumption that TF expression is a proxy for abundance and activity.

Reviewer #1 (Remarks to the Author):

This manuscript applied a recently proposed TWAS method to identify gene-trait associations that are potentially mediated through multi-omics of placenta, while accounting for distal eQTL. Significant gene-trait associations were identified, mostly for neonatal and metabolic traits. Follow-up studies show these significant gene-trait associations were enriched in cell growth and immunological pathways. The authors conducted comprehensive analyses to investigate these significant gene-trait associations. Overall, the findings are profound. But it is not clear what is the cohesive key contribution of this paper to our understanding of these investigated traits with respect to placental multi-omics variations. Here are my major concerns:

We are glad the reviewer found our analyses comprehensive and the results profound.

1. It is not clear how multi-omics data besides mRNA expression, like CpG methylation sites and miRNA expression data, were considered by MOSTWAS method.

This is an important point, and we have clarified this in two locations in the manuscript. First, in the Results section (*Multiple placental gene-trait associations detected across the life course* subsection; Page 6, Lines 7-11), we add:

“In this analysis, these regulatory biomarkers include potential regulatory protein (RP) encoding genes (as curated by TFcheckpoint³³), miRNAs, and CpG methylation sites from the ELGAN Study. We assume that these RP genes, miRNAs, and genes and other regulatory features local to these CpG methylation sites have distal effects on the transcription of genes of interest and thus potentially mediate distal-eQTLs to the gene of interest (Methods).”

In the Methods section (*Gene expression models* subsection; Page 21, Lines 25 to Page 22 Line 1), we add:

“Our assumption here is that distal-eQTLs of a gene that are local to transcription factor-encoding genes, miRNAs, or regulatory features local to CpG methylation sites may be potentially mediated by cis-QTLs to these local features. This assumption has been employed by multiple studies previously to identify trans-eQTLs in multiple tissues¹⁰⁷⁻¹¹⁰.”

We also have included several references to previous trans-eQTL mapping papers that employ similar ideas, which we leverage in the MOSTWAS models (Pierce *et al* 2014, *PLOS Genetics*; Hawe *et al* 2020, *bioRxiv*; Yang *et al* 2017, *Genome Research*; Yang *et al* 2019, *bioRxiv*).

2. The number of ~2,994 significant gene models seems quite a small proportion comparing to the total ~20K genes. Would a different method for training the gene expression prediction model be able to provide a greater number of significant gene models, like TIGAR (PMCID: PMC6698804), PMR (<https://www.nature.com/articles/s41467-020-17668-6>).

We appreciate that the reviewer directs us to the TIGAR and PMR methods; both of these methods are interesting and valuable extensions to local-only TWAS methods. There are multiple points here, however.

First, after a liberal filtering of low count genes from the RNA-seq panel (due to difficulties in extracting RNA from the placenta and degradation), we were only left with 12,020 genes, far less than 20,000 genes. Second, we do believe that more predictive gene models could be built if we integrated all possible TWAS methods (BSLMM from FUSION, DPR from TIGAR, PMR, etc). However, our main goal here was to integrate the rich multiomics data from the ELGAN study to identify genetic variants distal to genes to incorporate into the TWAS framework. As we mention in the paper and the MOSTWAS manuscript (Bhattacharya *et al* 2021, *PLOS Genetics*), one of the advantages of this framework are the prioritization of functional hypotheses for distal regulation of gene-trait associations. Lastly, we would also

like to point out that the inclusion criterion for MOSTWAS is conservative. Unlike the TIGAR and PMR methods, we have two feature selection criterion: (1) SNP heritability of expression and (2) predictive power through cross-validation. As Cao *et al* shows (Cao *et al* 2021, *PLOS Genetics*), both SNP heritability of expression and predictive power are important in determining TWAS power. In our TWAS, we only consider those genes with significantly positive SNP heritability (estimated with GREML-LDMS and assessed using a likelihood ratio test) and McNemar's cross-validation adjusted- $R^2 \geq 0.01$. Only approximately 4,000 genes showed significantly positive SNP heritability in the ELGAN dataset, and we built highly predictive models for nearly 75% of these genes. We believe these feature selection steps are necessary, steps which are not included in the TIGAR and PMR pipelines.

3. Please state clearly which tool/method was used to estimate heritability in this manuscript and plot the 95% confidence intervals of the heritability estimates.

There are three heritability estimations that we conduct here. First, we estimate the SNP heritability of gene expression. We use GREML-LDMS of this estimation, as detailed in the Methods section (*Estimation of SNP heritability of gene expression* subsection; Page 21, Lines 13-20):

“Heritability using genotypes within 1 Megabase of the gene of interest and any prioritized distal loci was estimated using the GREML-LDMS method, proposed to estimate heritability by correction for bias in LD in estimated SNP-based heritability¹⁰⁵. Analysis was conducted using GCTA v1.93.1¹⁰⁶. Briefly, Yang *et al* shows that estimates of heritability are often biased if causal variants have a different minor allele frequency (MAF) spectrums or LD structures from variants used in analysis. They proposed an LD and MAF-stratified GREML analysis, where variants are stratified into groups by MAF and LD, and genetic relationship matrices (GRMs) from these variants in each group are jointly fit in a multi-component GREML analysis.”

Next, we estimate SNP heritability of each trait using LD score regression. The 95% confidence intervals for these estimates are included in Supplemental Figure S1. We include in the Results section (*Complex traits are genetically heritable and correlated* subsection; Page 5, Lines 13-16) the following sentence:

“To quantify the total genetic contribution to each trait and the genetic associations shared between traits, using linkage disequilibrium (LD) score regression with LD scores generated for individuals of European ancestry from the 1000 Genomes projects^{31,32}, we estimated the SNP heritability (h^2) and genetic correlation (r_g) of these traits, respectively (Supplemental Figure S1 and S2).”

In the Methods section (*GWAS summary statistics* subsection; Page 20, Lines 21-23), we also add:

“SNP heritability for each trait and genetic correlations for all pairwise combinations of traits were estimated using LD score regression with the European ancestry sample from the 1000 Genomes Project as a reference for LD scores^{31,32}.”

Lastly, we also use a modified approach to LD score regression to estimate placental GREx-mediated heritability of traits, as detailed in the FUSION and RHOGE papers, respectively (Gusev *et al* 2016, *Nature Genetics*; Mancuso *et al* 2017, *American Journal of Human Genetics*). We describe this approach in the Methods section (*Genetic heritability and correlation estimation* subsection; Page 24, Lines 5-16):

“At the genome-wide genetic level, we estimated the heritability of and genetic correlation between traits via summary statistics using LD score regression³¹. On the predicted expression level, we adopted approaches from Gusev *et al* and Mancuso *et al* to quantify the

heritability (h_{GE}^2) of and genetic correlations (ρ_{GE}) between traits at the predicted placental expression level^{17,28}. We assume that the expected χ^2 statistic under a complex trait is a linear function of the LD score³¹. The effect size of the LD score on the χ^2 is proportional to h_{GE}^2 :

$$E[\chi^2] = 1 + \left(\frac{N_T l}{M}\right) h_{GE}^2 + N_T a,$$

where N_T is the GWAS sample size, M is the number of genes, l is the LD scores for genes, and a is the effect of population structure. We estimated the LD scores of each gene by predicting expression in European samples of 1000 Genomes and computing the sample correlations and inferred h_{GE}^2 using ordinary least squares. We employed RHOGE to estimate and test for significant genetic correlations between traits at the predicted expression level²⁸.

The reviewer is right about adding a figure that shows the precision of these heritability measurements. The new Supplemental Figure S10 now shows Wald-type confidence intervals for h_{GE}^2 . We copy the figure below:

4. A heatmap with all pair-wise correlation test results would be better than the current Figure 3C plot.

We appreciate the reviewer's comment about alternative approaches to plotting figures. We do include this heatmap in the original Supplemental Figure S11, but it is difficult to distinguish the significant correlations between traits of different categories. We believe the interesting results here are these cross-category correlations, and hence we provide the forest plot in the main Figure 3C. We did try to plot these cross-category correlations in Figure 3C as a heatmap, but this approach was the clearest approach.

5. A phewas-like Manhattan plot with respect to each of the examined genes would provide a more complete view of the analysis results for the phenome-wide association study in early- and later-life traits.

We include the boxplots in the main Figure 4A mainly to prioritize organ systems and trait categories as a whole for these 9 genes that showed TWAS associations in 3 or more trait categories. However, we do provide PheWAS-like Miami plots for this phenome-wide scan in UKBB in the original Supplemental Figure S12 for 4 genes that showed multiple enrichments across organ systems: *ATPAF2*, *IDI1*, *RPS25*, and *SEC11A*.

6. Please state if the LD matrix $\Sigma_{(S,S)}$ is the genotype correlation matrix or covariance matrix. Actually, for using the Z-score statistic as mentioned in the Methods for TWAS, the genotype correlation matrix should be used.

We appreciate this comment from the reviewer – it helps with the reader’s understanding, especially if they are not familiar with statistical genetic analyses. The reviewer is right that the LD matrix in the weight burden test is the genotype correlation matrix, as mentioned in the Gusev *et al* 2016 FUSION paper. We have now added to the Methods section (*Overall TWAS test* subsection; Page 23, Lines 11-13) the following sentence:

“The vector w_G represents the vector of SNP-gene effects from MeTWAS or DePMA and $\Sigma_{s,s}$ is the LD matrix (correlation matrix between genotypes) between the SNPs represented in w_G .”

Reviewer #2 (Remarks to the Author):

Bhattacharya et al reported a comprehensive study integrating placenta molecular QTLs and summary level results of 40 GWASes. This work applied existing methods (e.g. TWAS) and detected 248 gene-trait associations (GTA). The authors interpret the observation as placental genomic regulation impacts developmental programming across the life course and increase risk of many diseases. The paper is well-organized and presenting high quality data. I have the following concerns.

We appreciate that the reviewer found our paper well-organized and our data and results high quality.

1) A key conclusion of this paper is placental genomic regulation impacts developmental programming across the life course and disease risk, and such conclusion is based on the GTA detected by TWAS. However, the association between a pair of SNP and molecular trait, ie., molQTLs (e.g. expressional QTLs [eQTLs] or methylation QTLs [mQTLs]), could exist in multiple tissues. The data employed in the paper did not show the molQTLs underlying GTA were specific in placenta, therefore, caution will be needed in drawing the conclusion.

The reviewer is correct that our study is specific to the placenta, and thus more caution should be taken when presenting conclusions. As we now address horizontal pleiotropy (see response to Comment 2) in our study, we have replaced the first limitation in the Discussion section (Page 17, Lines 9-13) with the following:

“First, our analysis considers only placental tissue. Though many of our GTAs leverage distal-eQTL architecture which tend to be tissue-specific, the QTLs we leverage in TWAS may not be placenta-specific. A similar analysis across developmental and adult tissues could reveal more widespread genetic signals associated with these traits.”

2) In Discussion, it is mentioned, “instances of horizontal SNP pleiotropy, where SNPs influence the trait and expression independently, were not examined here.” But why? There are existing methods, such as coloc and eCAVIAR, to evaluate the probability that the placenta molecular trait and disease risk were controlled by the same variants. The data to apply these methods are basically the same as in TWAS.

The reviewer makes an important point here. We note that colocalization methods like coloc and eCAVIAR are not built to test for horizontal pleiotropy, but rather coincidence of genetic associations between two traits for variants. As Wainberg et al points out (Wainberg et al 2019, *Nature Genetics*), colocalization is an alternative to TWAS in gene prioritization for GWAS signal. A good figure that summarizes this is Figure 2 from the MRLink paper (van der Graaf et al 2020, *Nature Communications*); colocalization can only address the situation where the outcome is affected through two pathways with two different, correlated SNPs (or GRex of genes). Here, we are interested in the situation where the outcome is affected through two pathways from the same SNP/set of SNPs, where one pathway is unobserved. To test the degree of horizontal pleiotropy, we have now included an analysis using PMR-Egger (Yuan et al 2020, *Nature Communications*). Here, we test the null hypothesis that the horizontal pleiotropic effect is zero (see Equations 1-4 of Yuan et al) and report these results in Supplemental Table S6. We now add to the Results section (*Multiple placental gene-trait associations detected across the life course* subsection; Page 7, Line 26 to Page 8, Line 5):

“Next, we tested for horizontal pleiotropic effects of the SNPs employed in the models for TWAS-prioritized genes; if SNPs affect the outcome through a pathway independent of expression of the gene, the TWAS association may be biased^{37,38}. Here, using PMR-Summary-Egger³⁸, we test the magnitude of this null hypothesis for each of the 248 TWAS-prioritized GTAs. At FDR-adjusted $P < 0.05$, only three GTAs showed significant horizontal pleiotropic effects: *MOV10*, *SLC35G2*, and *HLA-A*, all associated with adult waist-hip ratio (Supplemental Table S6). These three genes may have upwardly biased TWAS associations, as the SNPs

used to construct their GReX may influence the outcome through a different molecular pathway.”

Accordingly, we remove our first limitation, which we replace with the caution outlined in the response to Comment 1.

3) TWAS was applied on GWAS summary data of 40 diseases (Table S1 and S2). Why limit to these 40 diseases? Since it is not clear how authors determined diseases other than these 40 were not relevant to placenta and TWAS was unlikely to detect TGA.

The reviewer is correct in pointing out that the DOHaD hypothesis can be applied to many more traits than just the 40 traits we study in our analysis. We started with 5 broad categories of traits (autoimmune, metabolic, cardiovascular, early childhood, and neuropsychiatric) that have particular relevance to the placenta from previous genomic and morphological analyses (References 1-8). From here, we identified large-scale GWAS initiatives that studied traits that fall in these categories and included a large, representative sample of traits in our analysis. Our work can be easily extended to other traits that were not studied in our paper. An important part of this work is the placental expression models we openly provide to the community to run TWAS in other GWAS of interest, accessible from Zenodo. The ELGAN DOHaD Atlas (<https://elgan-twas.shinyapps.io/dohad/>) also enables researchers to submit TWAS summary statistics, so results from other traits may be consolidated in one location for the community to browse and investigate.

4) The authors suggest “differing eQTL architectures in the datasets [ELGAN and RICHS] and different inclusion criteria for significant gene expression models”. It would be informative to systematically compare the eQTLs of these two placenta datasets. The difference in ELGAN and RICHS eQTLs could be of scientific importance reflecting the preterm status.

The reviewer poses an interesting question that is worth exploring in a separate paper. We believe that the differences in genetic effects on placental transcriptomics with pre-term status using both the ELGAN and RICHS data will be an important paper for the field and could reveal effect modifications on these QTLs by variables that are correlated with pre-term status. We are currently planning an extensive mega-analysis across different placental eQTL cohorts to map genome-wide QTLs, but several pre-processing steps need to be taken first (i.e., impute genetic data to the same reference, same pre-processing steps for RNA-seq data, isoform-level expression estimation with inferential replicates, percent spliced in estimation, etc). This analysis would be beyond the scope of this paper, as it is not the focus of our paper and goes beyond our data use agreement for the pre-processed RICHS data. Without aligning data from these cohorts, interpretability of differences in QTLs will be difficult. However, for added transparency, we expand the demographic and clinical summary statistic table in Supplemental Table S4 to both ELGAN and RICHS to highlight the difference in the cohorts. We also add at in the Results section (*Multiple placental gene-trait associations detected across the life course* subsection; Page 6, Lines 24-26):

“Summary statistics of demographic and clinical variables for the RICHS show similar distributions of race, though RICHS excluded all pre-term babies, a clear difference in these two cohorts (Supplemental Table S4).”

We also add in the Discussion section the following sentence to highlight the importance of this future analysis (Page 17, Lines 16-17):

“An extensive comparison of genome-wide eQTL architecture between ELGAN and RICHS, highlighting differences in genetic effects on gene expression across pre-term status, could be of particular scientific importance.”

5) Does HMDP has placenta expression data, so TGA detected by TWAS in human can be replicated in mice? If HMDP has placenta expression data, the methods and results section should provide more details.

The reviewer asks a relevant question. Unfortunately, HMDP does not have placental expression data from mice, which makes direct tissue-specific comparisons difficult. However, cis-eQTLs are employed in the predictive models MOSTWAS builds, and thus many of these are conserved across tissues. We use this idea to compare results from our human placenta TWAS with the results in other tissues from HMDP. We have adjusted the Results and Methods sections to mention that placental expression is not available in the HMDP. The Results section (*Body size and metabolic placental GTAs show trait associations in mice*; Page 12, Lines 3-4) now reads:

“This panel includes 100 inbred mice strains with extensive collection of obesity-related phenotypes from over 12,000 genes, with expression measured in a variety of adult tissues.”

Later in that same paragraph, we write (Lines 12-14):

“Though generalizing these functional results from non-placental tissue in mice to humans is tenuous, we believe these 36 individually significant genes in the HMDP are fruitful targets for follow-up studies.”

We add a new Methods section (*Human Mouse Diversity Panel*; Page 25, Lines 16-23), which reads:

“To provide some functional evidence of gene associations with metabolic traits, we evaluated the 109 metabolic trait-associated genes from our human placental TWAS in the Hybrid Mouse Diversity Panel (HMDP) for correlations with obesity-related traits in mice⁴⁷. This panel includes 100 inbred mice strains with extensive collection of obesity-related phenotypes (e.g., cholesterol, body fat percentage, insulin, etc) from over 12,000 genes, with expression measured in a variety of adult tissues (liver, adipose, aorta). We note that the HMDP only considers adult tissues and does not include placental gene expression. In the HMDP, we consider both trait correlation to tissue-specific gene expression and cis-GReX (genetically-regulated expression controlled by cis-eQTLs).”

In the third paragraph of the Discussion (Page 17, Lines 3-5), we also mention that generalizing to humans from mouse models is tenuous:

“Although these cis-GReX correlations from HMDP cannot be generalized from mice to humans, our in vitro assay provides valuable evidence for EPS15 genomic regulation in the placenta.”

Reviewer #3 (Remarks to the Author):

The overall goal of this paper was to characterize placental gene-trait associations relevant to the DOHAD hypothesis using the newly developed MOSTWAS method for eQTL analysis. The major strengths of this paper are the robust transcriptomics data and efforts to integrate this data using novel computational approaches. This paper is significant as it provides evidence of changes in placental gene expression related to genetic polymorphisms associated with metabolic and childhood/neonatal traits, supporting the DOHAD hypotheses. Overall, this paper is very ambitious and tackles many aspects of genetic control of placental gene expression, but is challenging to follow due to the sheer number of analyses being performed (RHOGE, MOSTWAS, FOCUS, GBAT, Mouse Diversity Panel, GREML-LDMS, MatrixQTL...). Overall, It would be helpful if the authors could go through and further clarify the goal's of each analysis. There are some additional areas that could be further explained in this paper, which are highlighted in the following comments:

We appreciate that the reviewer finds our paper significant and ambitious. We understand that the paper may be difficult given the numerous analyses included. Accordingly, at the reviewer's request, we have included an initial subsection in the Results section that outlines the different steps of the analysis with motivations for each leg of the analysis. We have also split Figure 1 into two figures to allow for more focus on the analytic pipeline in the new Figure 2. In the Methods sections, we have ensured that we include motivations for each analytic choice in the pipeline.

1. Demographic data on the ELGAN cohort is presented in Supplemental Table S4, but there is no demographic data provided on the RICHs cohort, which was used to validate the findings of gene expression prediction. How do the RICHs and ELGAN cohorts compare in terms of race, ethnicity, SES, rates of premature birth, and other key perinatal outcomes? It would be helpful to mention in the discussion about the differences in these cohorts and the validity of the RICHs cohort as an independent validation cohort.

We appreciate the reviewer asks about differences between the RICHs and ELGAN datasets. Supplemental Table S4 now includes a summary of more demographic and clinical variables about each dataset. Distributions of race are roughly similar across these two groups, though there is one significant difference. The RICHs excluded all pre-term births, whereas the ELGAN study includes only pre-term births; this results in a large difference in mean gestational age. This is a major difference, which we now highlight in the Results section (*Multiple placental gene-trait associations detected across the life course* section; Page 6, Lines 24-26):

“Summary statistics of demographic and clinical variables for the RICHs show similar distributions of race, though RICHs excluded all pre-term babies, a clear difference in these two cohorts (Supplemental Table S4).”

2. Did the RNA sequencing data from the RICHs cohort undergo the same preprocessing steps as the ELGAN cohort? (such as RUV-seq) This information is not described in the methods.

This is an important point raised by the reviewer. We received pre-processed expression data from the RICHs. We have included a reference to the paper that describe these pre-processing steps. For clarification, these pre-processing steps are different from those used in this analysis for the ELGAN data, and, in the Methods section (*Expression data* subsection), we add the following to make sure the reader is aware of differences in pre-processing:

“We obtained pre-processed RNA expression data from the Rhode Island Children's Health Study, as described before³⁶. Pre-processing steps for RNA expression data from the RICHs are different from those employed here in the ELGAN study.”

3. How did the authors handle cross-reactive/polymorphic probes in their analysis? These probes introduce bias in studies utilizing Illumina EPIC arrays (doi: <https://doi.org/10.1093/nargab/lqaa105>) and are frequently removed prior to downstream analyses.

We thank the reviewer for pointing out this salient step in pre-processing methylation data. The software included with the paper cited by the reviewer seems to have some dependency issues. However, we use a different annotation to account for cross-reactive/polymorphic probes on the EPIC chip. Using the catalogue from the maxprobes R package from Max Chen (Pidsley *et al* 2016, *Genome Biology*; McCartney *et al* 2016, *Genomics Data*). Of the 1,072 CpG sites we leverage to train predictive models, only 52 are cross-reactive/polymorphic. Subsetting to those genes with TWAS associations, we found 4 cross-reactive CpGs that were prioritized as a mediator for the GTA. We re-estimated models for these genes, along with the TWAS test of association and distal-SNPs added last test. Here is a summary of the changes to results:

- **cg11299304 was tagged as a predicted mediator of the *ZNF850* association with adult-onset asthma. After re-fitting, the overall TWAS Z-score remained in the same direction (new Z = 7.51) and the distal Z-score remained in the same direction (new distal Z = 6.21)**
- **cg08856764 was tagged as a predicted mediator of the *PLA2G2A* association with allergic disease. After re-fitting, the overall TWAS Z-score remained in the same direction (new Z = -5.89) and the distal Z-score remained in the same direction (new distal Z = -3.24)**
- **cg07424785 was tagged as a predicted mediator of the *PRRC2A* association with head circumference. After re-fitting, the overall TWAS Z-score remained in the same direction (new Z = 5.12) and the distal Z-score remained in the same direction (new distal Z = 2.20)**
- **cg21948827 was tagged as a predicted mediator of the *ZNF264* association with late puberty growth. This CpG site was the only predicted mediator for this TWAS association. The overall TWAS Z-score remained in the same direction (new TWAS Z = -5.21). However, we can no longer prioritize a distal functional hypothesis for this gene.**

Throughout the Results section and Supplemental Table S6, we have updated the results to reflect these 4 changes. In the Methods section (Page 22, Lines 1-2), we mention that we filter out cross-reactive probes for these TWAS-associated genes:

“For CpG methylation sites, we used the maxprobes R package to filter out cross-reactive or polymorphic probes, which may induce bias^{111–113}.”

4. How were the X and Y chromosome handled in this analysis?

This is an important clarification from the reviewer. Due to the difficulties in quality control for genetic data from sex chromosomes, we restrict our study to genetic variants, genes, CpG sites, or miRNAs on autosomes. We have added to the Methods section (*Data acquisition and quality control* subsection) that all genetic, transcriptomic, and methylomic data considered in this analysis are from autosomes.

5. How did the authors define transcription factors for MOSTWAS?

We appreciate the reviewer for raising this question, since we omitted a citation to TFcheckpoint as our source of potential transcription factors (Chawla *et al* 2013, *Bioinformatics*). This curated compendium included a set of 3,479 potential transcription factors with at least one experimental or computational source. Of these transcription factors, we were able to study 1,962 of them in the ELGAN RNA-seq panel. We contend that many of these 3,479 transcription factors from TFcheckpoint are not represented in the more recent

databases that the reviewer mentions in their Comment 9. However, we believe that there has been some evidence that these transcription factors may be involved in regulatory processes. Previous studies that conduct distal-eQTL mapping using mediation through cis-eQTLs use all genes in the transcriptome as potential mediators (Pierce *et al* 2014, *PLOS Genetics*; Hawe *et al* 2020, *bioRxiv*; Yang *et al* 2017, *Genome Research*; Yang *et al* 2019, *bioRxiv*). In an attempt to add some biological interpretation, we restrict to this superset of genes coding for regulatory proe.

Accordingly, we have added a citation to TFcheckpoint and, to soften our language, call the genes we consider as mediators “potential regulatory protein (RP)-encoding genes.” Specifically, in the Results section (*Multiple placental gene-trait associations detected across the life course* subsection; Page 6, Lines 6-11), we now write:

“In this analysis, these regulatory biomarkers include potential regulatory protein (RP) encoding genes (as curated by TFcheckpoint³³), miRNAs, and CpG methylation sites from the ELGAN Study. we assume that these RP genes, miRNAs, and genes and other regulatory features local to these CpG methylation sites have distal effects on the transcription of genes of interest and thus potentially mediate distal-eQTLs to the gene of interest (Methods).”

6. The authors address cellular heterogeneity present in placental omics data by adjusting for unwanted biological variation using RUVSeq. However, the authors do not adjust for this same variability in the DNA methylation data, and there are established methods to do this in methylation data (Houseman *et al*, 2015, Yuan *et al*, 2020)

We thank the reviewer for pointing this out. Our CpG methylation data does include an adjustment for cellular heterogeneity using surrogate variable analysis (Leek *et al* 2007, *PLOS Genetics*). We have previously used this adjustment in previous studies that used this methylomic data from ELGAN (Santos and Bhattacharya *et al* 2020, *Molecular Autism*; Santos *et al* 2019, *Epigenetics*). In the Methods section (*Methylation data* subsection; Page 20, Lines 6-9), we now write:

“Lastly, the ComBat function was used from the sva package to adjust for batch effects from sample plate⁹⁸. In addition, to account for cell-type heterogeneity, 5 surrogate values were estimated and removed from the data to account using the sva package, as previously described^{15,90,98}.”

7. For the heritable traits, can you provide some clarity if the GWAS results (HDL, Glucose, Cholesterol) that directly related to blood markers/metabolites were quantified in adults or children in the studies which they were derived? Also, the way this is presented may make it seem to a casual reader that these measurements were collected in the same population based on the labeling in some of the tables/figures (such as supplemental figure S9) and some of the wording (For example page 8 line 27-page 9 line 1). Readability could be improved by slight rewording.

We appreciate the reviewer in providing these comments that aid the readability of our manuscript. Unless otherwise indicated, all traits are measured in adults. Only those traits that fall in the neonatal/childhood outcomes category are measured in infants or children. We have added, throughout the Results section, multiple references to whether a trait is measured in children or adults to aid with readability. In addition, the first paragraph of the *Complex traits are genetically heritable and correlated* subsection (Page 5, Lines 1-11) now reads:

“We curated GWAS summary statistics from subjects of European ancestry for 40 non-communicable traits and disorders across five health categories to identify potential links to genetically-regulated placental expression (traits and cohorts for each GWAS are summarized in Supplemental Table S1, sample sizes are provided in Supplemental Table S2). These five categories of traits (autoimmune/autoreactive disorders, metabolic traits, cardiovascular disorders, early childhood outcomes, and neuropsychiatric traits) have

been linked previously to placental and fetal biology and morphology¹⁻⁸. These 40 traits, derived from 5 different consortia (Supplemental Table S1), comprise of 3 autoimmune/autoreactive disorders, 8 body size/metabolic traits, 4 cardiovascular disorders, 14 neonatal/early childhood traits, and 11 neuropsychiatric traits/disorders²³⁻²⁷. The 26 traits that are not categorized as neonatal/early childhood traits are measured exclusively in adults. In addition, these 40 GWAS are not derived from the same samples of patients.”

We believe that these modifications significantly help with the flow of the paper.

How was the threshold of adjusted $R^2 > 0.01$ selected? These values of the cross validation and out of sample R^2 seem low-is there an example of another paper that uses this threshold? Also, in Page 5 Line 25, the authors refer to an “adjusted” R^2 -How was this adjusted?

This is a great clarification question. $R^2 \geq 0.01$ is the traditional cutoff used in the original TWAS papers (Gamazon *et al* 2015, *Nature Genetics*; Gusev *et al* 2016, *Nature Genetics*), corresponding to a correlation of 0.10 between the observed expression values and the predicted genetically-regulated expression values. We have cited multiple other TWAS papers that use this $R^2 \geq 0.01$ cutoff. In a majority of TWAS papers, only cross-validation $R^2 \geq 0.01$ is used as a feature selection step for the predictive power of the models, given the dearth of large enough tissue-specific eQTL datasets. We provide out-of-sample prediction estimates to show relatively strong portability of models across two datasets, especially since MOSTWAS is a relatively new method. In addition, MOSTWAS considers the McNemar’s adjusted R^2 , instead of a traditional R^2 (squared correlation), which adjusts the R^2 for sample size. In our experience, feature selecting on this adjusted R^2 leads to more portable models, which is important when conducting the TWAS test of association. We now define the McNemar’s adjusted R^2 mathematically in the Methods section (*Gene expression models* subsection; Page 22, Lines 13-19):

“We considered only genes with significantly positive heritability at nominal $P < 0.05$ using a likelihood ratio test and five-fold McNemar’s adjusted cross-validation $R^2 \geq 0.01$, a cross-validation cutoff used by many previous TWAS analyses^{16,17,28,36,75,114,115}. McNemar’s adjustment to the traditional R^2 is computed as

$$R_{adjusted}^2 = 1 - (1 - R^2) \frac{(n - 1)}{(n - v - 1)},$$

where n is the sample size and v is the number of predictors in this linear model. Since this R^2 is computed only between the observed and predicted expression values, $v = 1$.”

8. The authors perform in vitro validation using JEG-3 Cells, which are not immortalized trophoblast cell lines. The JEG-3 cells are derived from a choriocarcinoma cell line that is used to model the placental trophoblast because they synthesize placental hormones and express key placental hormones including HLA-G. There are other cell lines, such as SWAN-71 and HTR8-SV/neo cells, which are derived from immortalized trophoblast cells. Thus, this cell line should be referred to appropriately in the manuscript text. What was the justification for using this particular cell line?

JEG-3 cells are indeed choriocarcinoma epithelial cells derived from the placenta, and exhibit features that prove their fit for this experiment. This correction has been added to the manuscript in the Methods section describing cell culture techniques. Specifically, the revised manuscript now states (Page 25, Lines 27-28):

“The JEG-3 choriocarcinoma cells were purchased from the American Type Culture Collection.”

JEGs exhibit first trimester-like phenotypes with regards to steroidogenesis, including the synthesis and secretion of hCG, human placenta lactogen, progesterone, estrone, and estradiol. The revised manuscript now states, in the Results section (*In-vitro assays reveal widespread transcriptomic consequences of EPS15 knockdown* subsection; Page 14, Lines 19-21):

“JEG-3 cells were selected for study based on their know first trimester-like phenotypes, including the synthesis and secretion of hCG, human placenta lactogen, progesterone, estrone, and estradiol.”

Immortalized cell-lines generally demonstrate an absence of hormone secretion. JEG-3 cells are widely used to study molecular mechanisms underlying proliferation and gene expression, whereas other cell lines derived from immortalized trophoblast cells such as SV/neo cells may be better suited for studies relating to invasion and migration.

9. Can the authors comment more on the logic of selecting the gene EPS15 for experimental validation, out of the 9 TFs they identified? The authors state that this gene is highly expressed in the placenta, but the citation they reference (#45 <https://doi.org/10.1093/humup/dmq052>) only mentions EPS15 once in a table of imprinted genes, using data derived from an imprinted gene catalogue (www.geneimprint.com). The fact that it is imprinted raises additional questions about how this is incorporated in the results. From what I can discern, EPS15 is a substrate for the EGFR receptor, and has been shown to localize within the nucleus ([doi:10.1186/1478-811X-7-24](https://doi.org/10.1186/1478-811X-7-24)), but I cannot find any literature directly implicating it as a transcription factor. Moreover, it is not included in commonly utilized database of transcription factors (<https://doi.org/10.1016/j.cell.2018.01.029>, <https://doi.org/10.1038/s41588-019-0411-1>). More insight into the function of this gene in the placenta and its role in transcriptional regulation (overall) is warranted, as it was highlighted as one of the main findings in the abstract.

As we mention in our response to the reviewer’s Comment 5, we use the TFcheckpoint database that combined multiple experimental and computational databases of potential transcription factors to identify a large set of potential transcription factors or regulatory proteins. EPS15 has been implicated as a potential transcription factor by GOC (Carbon *et al* 2009, *Bioinformatics*) and TFCat (Fulton *et al* 2009, *Genome Biology*). We subject our functional hypotheses of distal regulation to multiple computational tests across datasets. First, rigorous mediation analysis is used during the MOSTWAS gene expression model building to identify *EPS15* as a potential mediator of the distal-eQTL relationship between genetic variants around *EPS15* and *SPATA13/FAM214A*. Second, we use the distal-added last test to implicate the SNP to *EPS15* to *SPATA13/FAM214A* relationships in the associations with waist-hip ratio. Third, we use the RICHs data to run two analyses that reveal directional associations between the genetically-regulated expression of *EPS15* and *FAM214A*: gene-based association testing of distal-eQTLs (Liu *et al* 2020, *Genome Biology*) and Mendelian Randomization (Burgess and Thompson 2017, *European Journal of Epidemiology*). By observing evidence from 3 rigorous computational analysis across different datasets, we prioritized the experimental study of the transcriptomic consequences of *EPS15* in placenta-derived trophoblasts.

We did not originally speculate on the function of *EPS15*, but there are several salient functions that have been identified in literature. We now add, in the Discussion section, more context on the suggesting roles of *EPS15* in transcription regulation and the potential function of *EPS15* in the placenta. We paste this new paragraph below (Page 16, Lines 8-23):

“MOSTWAS also generates hypotheses for regulation of TWAS-detected genes, through distal mediating biomarkers, like transcription factors, miRNAs, or products downstream of CpG methylation islands²². Our computational results prioritized 89 GTAs with strong distal associations. We interrogated one such functional hypothesis: *EPS15*, a predicted RP-encoding gene in the EGFR pathway, regulates two TWAS genes positively associated with waist-hip ratio - *FAM214A*, a gene of unknown function, and *SPATA13*, a gene that regulates

cell migration and adhesion⁶⁰⁻⁶². In fact, *EPS15* itself showed a negative TWAS association with waist-hip ratio. In particular, *EPS15*, mainly involved in endocytosis, is a maternally imprinted gene and predicted to promote offspring health^{49,63-65}. There is ample literature that implicates the protein product of *EPS15* as a direct or indirect transcription regulator. The protein Eps15 is an adaptor protein that regulates intracellular trafficking and has been detected in the nucleus of mammalian cells⁶⁶. Once in the nucleus, Eps15 has shown to positively modulate transcription in a GAL4 transactivation assay⁶⁷. Furthermore, Eps15 and its binding partner intersectin activate the Elk-1 transcription factor, pointing to Eps15's function in regulating gene expression in the nucleus⁶⁸. Specific to the placenta, it has been proposed, through mouse models, that Eps15's interactions with multiple proteins suggest a role in cell adhesion of trophoblast to endothelial cells through biogenesis of exosomes and extracellular vesicles, a critical part of placental and fetal development⁶⁹⁻⁷¹.

We also soften some language in our Discussion and emphasize to the reader that our results cannot implicate a direct causal effect. We now write this in the Discussion section (Page 16, Line 27 to Page 17, Line 1):

“Though not implicating a direct causal effect, *EPS15*'s inverse association with *SPATA13* and *FAM214A* could provide more context to its full influence in placental developmental programming, perhaps by affecting cell proliferation or adhesion pathways.”

10. The authors describe experiments where *EPS15* knockdown causes increased expression of genes *SPATA13* and *FAM214A* (Page 13, Line 2, and in Figure 5B). This would indicate that this gene is a negative regulator since less activity of this TF is leading to increased expression. However, in supplemental figure 13, they show positive correlations between *EPS15* and *SPATA13* and *FAM214A* in the RICHs cohort, which would implicate it as a positive regulator (Increased TF expression, Increased expression of downstream genes) So the data they are presenting here appears contradictory. Can the authors explain this?

We appreciate that the reviewer asks this important question, as it is true that the results in the old Supplemental Figure S13 appear contradictory to those shown in Figure 5. Supplemental Figure S13 simply provides correlations between the full placental expression of mediating genes and TWAS-identified genes, which we include as a description of the data at face value. We then dig deeper using more sophisticated computational methods: the associations shown in Figure 4B (association between placental GReX of RP expression and placental expression of TWAS gene) and Figure 4C (causal effect estimate of RP expression on placental expression of TWAS gene) both use genetics as a causal anchor. We prioritize *EPS15* for the *in vitro* experiments due to these computational results shown in Figure 4, that use associations between genetically-regulated expression of *EPS15* or genetic variants as instrumental variables. The correlations presented in Supplemental Figure S13 reflect total expression of these genes, subject to multiple post-transcriptional processes which may induce a different correlation than those with GReX. In the interest of thoroughness, we have elected to discuss this phenomenon quickly at the end of the second paragraph in the *MOSTWAS reveals functional hypotheses for distal placental regulation of GTAs* subsection (Page 13, Lines 24-27), where we write:

“However, as discussed in previous TWAS and MR studies^{17,53}, correlations between GReX and a phenotype are not equivalent to correlations between full expression and the phenotype, as full expression is subject multiple post-transcriptional process, while GReX is not.”

We also soften some language in our Discussion and emphasize to the reader that our results cannot implicate a direct causal effect. We now write this in the Discussion section (Page 16, Line 27 to Page 17, Line 1):

“Though not implicating a direct causal effect, *EPS15*'s inverse association with *SPATA13* and *FAM214A* could provide more context to its full influence in placental developmental programming, perhaps by affecting cell proliferation or adhesion pathways.”

11. In the methods, the authors describe confirming gene expression changes in JEG-3 cells through RT-PCR (Page 23, Lines 4-5). However, it appears later in the methods (Page 23, Lines 13-16) that differential gene expression was performed, which is presented in the results and in Figure 5B. However, in the methods there isn't any information provided about how RNA was isolated from the JEG-3 cells, or any details about the sequencing (i.e library prep method, sequencer, QC). Please provide this information.

The following information has been added to the revised manuscript in the Methods section (*mRNA expression by quantitative Real-Time Polymerase Chain Reaction and RNA sequencing* subsection). The revised manuscript now states (Page 26, Line 24 to Page 27, Line 2):

“Treated and untreated samples of JEG-3 RNA previously extracted using the AllPrep DNA/RNA/miRNA Universal Kit were submitted to the High Throughput Sequencing Facility at UNC Chapel Hill for RNA sequencing. Total RNA samples were submitted for sequencing using the HS4000 HO platform. Libraries were prepped with the Kapa Stranded mRNA-Seq kit from Illumina Platforms. Sequencing was performed after all samples passed QAQC, with a paired-end read type, with a read length of 2x75.”

12. As a follow up to this, the authors plated cells in duplicate, then performed siRNA isolation in duplicate. For the real time PCR, the samples were also run in technical duplicate (Page 23, line 2). The authors then had 3 degrees of freedom for their biological and technical duplicates (Page 23 Line 10) for pairwise T tests. Can the authors describe in more detail how the technical and biological duplicates were used to calculate the fold change using the Delta Delta CT method. Was this calculated separately for each sample? Also, if the authors also did RNA sequencing, was this performed only in duplicate? Normally the minimum number of samples presented in RNA sequencing experiments with in vitro samples are triplicates. (Example: <https://doi.org/10.1111/aji.12722>)

Samples were prepared in both biological and technical duplicate. Fold change calculations using the Delta Delta CT method was calculated for each sample individually. The revised manuscript now states in the Methods section (*mRNA expression by quantitative Real-Time Polymerase Chain Reaction and RNA Sequencing* subsection; Page 26, Lines 18-22):

**“Each sample was prepared in biological duplicate and technical duplicate. These samples were pooled together for sequencing to yield data representing four samples per exposure group. Fold change calculations using the Delta Delta CT method was calculated for each sample individually:
Delta CT_{treated} = CT_{GOI, treated} – CT_{House, treated}.”**

RNA sequencing was performed in duplicate, as we were limited by the reagent availability. This information has been incorporated into the Methods section (*mRNA expression by quantitative Real-Time Polymerase Chain Reaction and RNA sequencing* subsection; Page 26, Line 24 to Page 27, Line 2):

“Treated and untreated samples of JEG-3 RNA previously extracted using the AllPrep DNA/RNA/miRNA Universal Kit were submitted to the High Throughput Sequencing Facility at UNC Chapel Hill for RNA sequencing. Total RNA samples were submitted for sequencing using the HS4000 HO platform. Samples were sequenced in duplicate, and libraries were prepped with the Kapa Stranded mRNA-Seq kit from Illumina Platforms. Sequencing was performed after all samples passed QAQC, with a paired-end read type, with a read length of 2x75.”

The reviewer is correct that 2v2 RNA-seq experiments cannot provide strong power for analysis. However, as Schurch *et al* shows in their benchmarking analysis (Schurch *et al* 2016, *RNA*),

DESeq2 and other differential expression analysis methods control false positive rates adequately even at low sample sizes (down to duplicate, shown in Figure 2). On the other hand, the true positive rate (percent of true positives actually estimated to be positive) greatly suffers at low sample sizes. Accordingly, we report a more stringent set of differentially expression genes: those with absolute \log_2 -fold change of 0.5 and P-value less than 1.32×10^{-6} , a strict Bonferroni correction across the 37,788 assayed genes. This P-value correction is far more conservative than the Benjamini-Hochberg procedure previously used. Despite the limited sample size, we believe that this RNA-seq experiment serves as an exploratory next step to our qRT-PCR analysis, to look at further transcriptomic consequences of *EPS15* knockdown. We have amended the Results section to include a caution about the small sample size, our strict definition of a differentially expressed gene, updated ontology enrichment results, and softened interpretation and conclusions. The last paragraph of the Results section (*In-vitro assays reveal widespread transcriptomic consequences of EPS15 knockdown* subsection; Page 15, Lines 3-18) now reads:

“Due to small sample sizes, we define a differentially expression gene with absolute \log_2 -fold change greater than 0.5 at $P < 1.32 \times 10^{-6}$, a Bonferroni correction across all assayed genes (Methods). We detected 694 genes down-regulated and 814 genes up-regulated in the *EPS15* knockdown cells, validating the negative correlations between *EPS15* and *SPATA13* and *FAM214A* observed in qRT-PCR (Figure 5B, Supplemental Table S16-S17). In particular, these down-regulated genes were enriched for cell cycle, cell proliferation, or replication ontologies, while up-regulated genes were enriched for multiple different pathways, including lipid-related processes, cell movement, and extracellular organization (Figure 5C, Supplemental Table S18-S19). Enrichments for cellular, molecular, and disease pathway ontologies support these enrichments (Supplemental Figure S14, Supplemental Table S18-S19). Though we could not study the effects of these three genes on body size-related traits, cis-GReX correlation analysis from the HMDP did reveal a negative cis-GReX correlation ($r = -0.31$, FDR-adjusted $P = 0.07$) between *Eps15* (mouse analog of human gene *EPS15*) and free fatty acids in mouse liver (Supplemental Table S11). These results prioritize *EPS15* for further study in larger cell line or animal studies as a potential regulator for multiple downstream genes, perhaps for genes affecting cell proliferation and replication in the placenta, like *SPATA13*⁶⁰.”

The Methods section (*Differential expression analysis* sub-subsection; Page 27, Lines 12-17) now reads:

“Although false positive rates are well-controlled even at low sample sizes¹²⁰, true positive rates at such a low sample size are low for smaller thresholds of log-transformed fold changes. Thus, guided by Schurch et al’s analysis, due to very limited sample size, we considered a gene to be differentially expressed if the absolute \log_2 -fold change is greater than 1 and $P < 0.05/37,788 = 1.32 \times 10^{-6}$. This P-value threshold is a strict Bonferroni threshold across 37,788 quantified genes.”

13. The authors include an analysis using the Hybrid Mouse diversity panel to look at correlations with obesity related traits. However, the authors should mention in the discussion the limitations of generalizing between mouse and human studies of the placenta, given that the biology is so different.

We agree with the reviewer that generalizations from mice to humans are difficult. We have already included this in our concluding statement of the *Body size and metabolic placental GTAs show trait associations in mice* subsection (Page 12, Lines 12-14):

“Though generalizing these functional results from non-placental tissue in mice to humans is tenuous, we believe these 36 individually significant genes in the HMDP are fruitful targets for follow-up studies.”

We also include a statement in the second paragraph of the Discussion section that emphasizes this important point (Page 17, Lines 3-5):

“Although these cis-GReX correlations cannot be generalized from mice to humans, our *in vitro* assay provides valuable evidence for *EPS15* genomic regulation in the placenta.”

14. What do the authors mean when they say “collection of negative control variables”-page 15, line 12.

We include this statement in the Discussion to point to current methodology that is under development in our group. Negative control variables are outcomes that are known not to be causally affected by the treatment of interest. They have been used in causal inference to rule out non-causal associations and decrease bias in associative studies. In this TWAS case, one can imagine using outcomes that are unlikely to be caused by gene expression (say, certain lifestyle choices) as an adjustment in the predictive models of gene expression. Accordingly, properly chosen negative controls should help in rooting out bias in TWAS tests of association. We have included two references to causal inference literature in this sentence for the reader (Page 17, Lines 18-20):

“An interesting future endeavor could include negative control variables to account for unmeasured confounders in predictive models to allow for more generalizability of predictive models^{73,74}.”

15. Moreover, in the discussion, the authors should discuss the assumptions made in this analysis, particularly the assumption that TF expression is a proxy for abundance and activity.

We appreciate this comment from the review and have included a few sentences in the Discussion about the assumptions we make about regulatory protein expression (Page 17, Lines 23-28), pasted below:

“Fifth, we curated a list of regulatory proteins to include as potential mediators but use RNA expression of the genes that code for these proteins as a proxy for abundance. We contend that RNA abundance of the gene is a noisy estimate of the protein abundance. An interesting extension of this analysis could consider a proteome-wide association study, using the MOSTWAS framework to identify protein interactions that are disease-related.”

Reviewers' Comments:

Reviewer #1:

Remarks to the Author:

The authors did a great job to address my comments. Conducting TWAS using both cis- and trans-eQTL information with the fetal tissue gene expression data is valuable to the field.

My only comment is how the authors would share their summary level eQTL weights trained by MOSTWAS and their TWAS summary statistics with the public?

Reviewer #2:

Remarks to the Author:

My questions and concerns are adequately addressed.

Reviewer #3:

Remarks to the Author:

The authors of this manuscript have done an excellent job adding details that have clarified the results of this manuscript. I appreciate the inclusion of previous trans-eqtl papers that integrate the MOSTWAS method. Necessary details have now been added to this most recent draft that address reviewer concerns, including the addition of polymorphic probes, and adjustment for cellular heterogeneity. I have a few minor comments to further improve the manuscript.

1. I appreciate the authors tempering their language regarding EPS15 as a TF, and instead referring to it as a regulator protein encoding gene. Figure 2 provides a great summary, however, in panel D they still refer to TF activity of EPS15. This is not in alignment with the tempered methods and results.

2. Consider relabeling the x axis in Figure 5A from group of traits to ICD code block

3. Line 191- there is a space missing between adult and BMI

4. The authors explain in the response to reviewers the rationale behind the selection of the JEG-3 cells. However, the limitations of the JEG-3 cells are not mentioned in the discussion of the paper. It is important for a broad range of readers that might access nature communications to understand that choriocarcinoma derived placental cells may not accurately reflect the placental transcriptome, but are a reasonable representation.

5. I understand that both of these cohorts have multiple papers explaining how samples were collected, but it may be good to briefly summarize this in the online methods. One key component is the exclusion/inclusion criteria. The samples are derived from different gestational age infants, but there may be other key differences in inclusion/exclusion that are not discussed.

6. Instead of saying the pre-processing steps are different, can the authors briefly outline the main differences. Would this impact their results?

November 21, 2021

Enclosed is a response to reviewers for the manuscript:

"Genetic control of fetal placental genomics contributes to development of health and disease."

We were glad to see that the reviewers found our study substantial, comprehensive, and ambitious with key contributions to genetic determinants of placental genomic regulation in the context of the Developmental Origins of Health and Disease hypothesis. Specifically, we thank all three reviewers for their thorough comments and suggestions that have greatly improved our paper.

Starting on the next page, we address each reviewer's comments point-by-point.

We thank the reviewers and the editorial board at *Nature Communications*.

Sincerely,

Arjun Bhattacharya, PhD
Postdoctoral Fellow
Department of Pathology and Laboratory Medicine
Institute for Quantitative and Computational Biosciences
David Geffen School of Medicine
University of California, Los Angeles

Rebecca C. Fry, PhD
Carol Remmer Angle Distinguished Professor and Associate Chair
Department of Environmental Sciences and Engineering
Director, Institute for Environmental Health Solutions
Director, UNC Superfund Research Program
Director, Graduate Studies, Curriculum in Toxicology
University of North Carolina at Chapel Hill

Hudson P. Santos, Jr., PhD, RN
Associate Professor
School of Nursing
Director, Biobehavioral Laboratory and HEalth Resilience and Omics Science (HEROS) Hub
Director, Training & Mentorship Division, Institute for Environmental Health Solutions
University of North Carolina at Chapel Hill

Reviewer #1 (Remarks to the Author):

The authors did a great job to address my comments. Conducting TWAS using both cis- and trans-eQTL information with the fetal tissue gene expression data is valuable to the field.

We thank the review for their helpful reviews that have improved our manuscript through revisions.

My only comment is how the authors would share their summary level eQTL weights trained by MOSTWAS and their TWAS summary statistics with the public?

The models are available through Zenodo, as we include in the Data Availability section:

“All models and full TWAS results can be accessed at <https://doi.org/10.5281/zenodo.4618036>¹²¹.”

Reviewer #2 (Remarks to the Author):

My questions and concerns are adequately addressed.

We thank the review for their helpful reviews that have improved the clarity of methods manuscript through revisions.

Reviewer #3 (Remarks to the Author):

The authors of this manuscript have done an excellent job adding details that have clarified the results of this manuscript. I appreciate the inclusion of previous trans-eqtl papers that integrate the MOSTWAS method. Necessary details have now been added to this most recent draft that address reviewer concerns, including the addition of polymorphic probes, and adjustment for cellular heterogeneity. I have a few minor comments to further improve the manuscript.

We thank the reviewer for all their helpful comments. We address their additional points below.

1. I appreciate the authors tempering their language regarding EPS15 as a TF, and instead referring to it as a regulator protein encoding gene. Figure 2 provides a great summary, however, in panel D they still refer to TF activity of EPS15. This is not in alignment with the tempered methods and results.

We thank the reviewer for pointing out this oversight. The corrected Figure 2 now refers to regulatory proteins, instead of transcription factors.

2. Consider relabeling the x axis in Figure 5A from group of traits to ICD code block

We thank the reviewer for this comment – we have kept the label in Figure 5A as “Group of trait” as these groups do not map one-to-one to ICD code blocks. We do add more detail in the legend to point out that these groups are generally grouped around ICD code blocks.

1. Line 191- there is a space missing between adult and BMI

Thank you for pointing this out. We have corrected this.

4. The authors explain in the response to reviewers the rationale behind the selection of the JEG-3 cells. However, the limitations of the JEG-3 cells are not mentioned in the discussion of the paper. It is important for a broad range of readers that might access nature communications to understand that choriocarcinoma derived placental cells may not accurately reflect the placental transcriptome, but are a reasonable representation.

The reviewer is correct in this request. We have included a sentence in the Discussion (Page 17, Line 5) that mentions the main limitation of using these JEG3 cells:

“JEG3 cells are reliable in use and provide accurate results when investigating specific cellular responses, such as the placental gene expression experiments used in this study; however, these cell lines do not capture interactions between cell types in the placental tissue and its effects on the placental transcriptome, as a whole.”

5. I understand that both of these cohorts have multiple papers explaining how samples were collected, but it may be good to briefly summarize this in the online methods. One key component is the exclusion/inclusion criteria. The samples are derived from different gestational age infants, but there may be other key differences in inclusion/exclusion that are not discussed.

This is a subtle but important point that we now include in the Methods section (Page 20):

*“Differences in inclusion/exclusion criteria between ELGAN and RICHS
We highlight some differences in inclusion and exclusion criteria employed by ELGAN and RICHS. ELGAN enrolled children born extremely preterm (less than 28 weeks gestation) and surviving 28 days postnatally, with full details of the study recruitment and descriptive statistics of the cohort in O’Shea et al 21. In contrast, as mentioned in Peng et al36, the RICHS sample consists of term infants (≥37 weeks gestation, not twins) born without serious pregnancy complications or congenital and chromosomal abnormalities. In addition, RICHS oversampled*

for large-for-gestational age (LGA, >90% 2013 Fenton Growth Curve) and small-for-gestational age (SGA, <10% 2013 Fenton Growth Curve) infants.”

6. Instead of saying the pre-processing steps are different, can the authors briefly outline the main differences. Would this impact their results?

We have included differences in pre-processing in the Methods section, with a brief discussion of how these differences may affect eQTL signal across two cohorts. The impact of differences in pre-processing and QC of RNA-seq data on distal-eQTL signal merits a full simulation and real data-based methodological study. Our expectation is that the local-eQTL signal (which is largely conserved across cells and tissues) will not be affected, given proper covariate selection. However, we hypothesize that distal-eQTLs will have key differences due to how different normalization and QC steps affect removal of noise attributable to cell-type heterogeneity. We include the following sentences in the Methods (Page 19, Line 19):

“Pre-processing steps for RNA expression data from the RICHS are different from those employed here in the ELGAN study (e.g., using EDASeq and edgeR for GC bias correction and normalization³⁵); differences in pre-processing may affect inferred distal-eQTL architecture, as cell-type heterogeneity is captured and removed differently across ELGAN and RICHS^{22,89,90}.”